# MimicGen: A Data Generation System for Scalable Robot Learning using Human Demonstrations

**Abstract:** Imitation learning from a large set of human demonstrations has proved to be an effective paradigm for building capable robot agents. However, the demonstrations can be extremely costly and time-consuming to collect. We introduce MimicGen, a system for automatically synthesizing large-scale, rich datasets from only a small number of human demonstrations by adapting them to new contexts. We use MimicGen to generate over 50K demonstrations across 18 tasks with diverse scene configurations, object instances, and robot arms from just ∼200 human demonstrations. We show that robot agents can be effectively trained on this generated dataset by imitation learning to achieve strong performance in long-horizon and high-precision tasks, such as multi-part assembly and coffee preparation, across broad initial state distributions. We further demonstrate that the effectiveness and utility of MimicGen data compare favorably to collecting additional human demonstrations, making it a powerful and economical approach towards scaling up robot learning. Videos and additional results at https://sites.google.com/view/corl2023mimicgen.

**Keywords:** Imitation Learning, Manipulation

## 1 Introduction

Imitation learning from human demonstrations has become an effective paradigm for training robots to perform a wide variety of manipulation behaviors. One popular approach is to have human operators teleoperate robot arms through different control interfaces [1, 2], resulting in several demonstrations of robots performing various manipulation tasks, and consequently to use the data to train the robots to perform these tasks on their own. Recent attempts have aimed to scale this paradigm by collecting more data with a larger group of human operators over a broader range of tasks [3–6]. These works have shown that imitation learning on large diverse datasets can produce impressive performance, allowing robots to generalize toward new objects and unseen tasks. This suggests that a critical step toward building generally capable robots is collecting large and rich datasets.

However, this success does not come without costly and time-consuming human labor. Consider a case study from robomimic [7], in which the agent is tasked with moving a coke can from one bin into another. This is a simple task involving a single scene, single object, and single robot; however, a relatively-large dataset of 200 demonstrations was required to achieve a modest success rate of 73.3%. Recent efforts at expanding to settings with diverse scenes and objects have required orders of magnitude larger datasets spanning tens of thousands of demonstrations. For example, [3] showed that a dataset of over 20,000 trajectories enables generalization to tasks with modest changes in objects and goals. The nearly 1.5-year data collection effort from RT-1 [5] spans several human operators, months, kitchens, and robot arms to produce policies that can rearrange, cleanup, and retrieve objects with a 97% success rate across a handful of kitchens. Yet it remains unclear how many years of data collection would be needed to deploy such a system to kitchens in the wild.

We raise the question — how much of this data actually contains unique manipulation behaviors? Large portions of these datasets may contain similar manipulation skills applied in different contexts or situations. For example, human operators may demonstrate very similar robot trajectories to grasp a mug, regardless of its location on one countertop or another. Re-purposing these

trajectories in new contexts can be a way to generate diverse data without much human effort. In fact, several recent works build on this intuition and propose imitation learning methods that replay previous human demonstrations [8–11] (**more related work discussion in Appendix D**).

While promising, these methods make assumptions about specific tasks and algorithms that limit their applicability. Instead, we seek to develop a general-purpose system that can be integrated seamlessly into existing imitation learning pipelines and improve the performance of a wide spectrum of tasks.

In this paper, we introduce a novel data collection system that uses a small set of human demonstrations to automatically generate large datasets across diverse scenes. Our system, **MimicGen**, takes a small number of human demonstrations and divides them into object-centric segments. Then, given a new scene with different object poses, it selects one of the human demonstrations, spatially transforms each of its object-centric segments, stitches them together, and has the robot follow this new trajectory to collect a new demonstration. While simple, we found

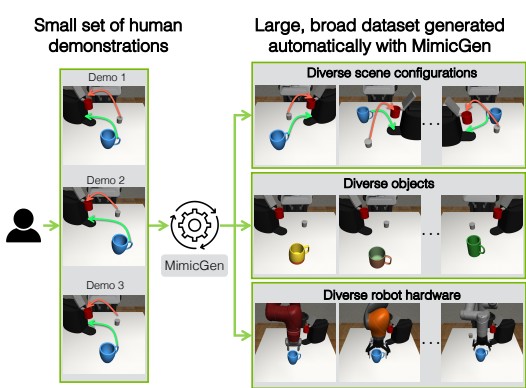

Figure 1: **MimicGen Overview.** We introduce a data generation system that can produce large diverse datasets from a small number of human demonstrations by re-purposing the demonstrations to make them applicable in new settings. We apply MimicGen to generate data across diverse scene configurations, objects, and robot hardware.

that this method is extremely effective at generating large datasets across diverse scenes and that the datasets can be used to train capable agents through imitation learning.

**We make the following contributions:**

• We introduce MimicGen, a system for generating large diverse datasets from a small number of human demonstrations by adapting the human demonstrations to novel settings.

• We demonstrate that MimicGen is able to generate high-quality data to train proficient agents via imitation learning across diverse scene configurations, object instances, and robot arms, all of which are unseen in the original demos (see Fig. 1). MimicGen is broadly applicable to a wide range of long-horizon and high-precision tasks that require different manipulation skills, such as pick-and-place, insertion, and interacting with articulated objects. We generated 50K+ new demonstrations for 18 tasks across 2 simulators and a physical robot arm using only ∼200 source human demos.

• Our approach compares favorably to the alternative of collecting more human demonstrations — using MimicGen to generate an equal amount of synthetic data (e.g. 200 demos generated from 10 human vs. 200 human demos) results in comparable agent performance — this raises important questions about when it is actually necessary to request additional data from a human.

## 2 Related Work

Some robot data collection efforts have employed trial-and-error [12–17] and pre-programmed demonstrators in simulation [18–22], but it can be difficult to scale these approaches to more complex tasks. One popular data source is human demonstrators that teleoperate robot arms [2–6,23–27], but collecting large datasets can require extensive human time, effort, and cost. Instead, MimicGen tries to make effective use of a small set of human samples to generate large datasets. We train policies from our generated data using imitation learning, which has been used extensively in prior work [1,19,25,28–34]. Some works have used offline data augmentation to increase the dataset size for learning policies [7,35–45] — in this work we generate new datasets online. Our data generation method employs a similar mechanism to replay-based imitation approaches [8–11, 46–48], which solve tasks by having the robot replay prior demonstrations. **More discussion in Appendix D.**

## 3 Problem Setup

**Imitation Learning.** We consider each robot manipulation task as a Markov Decision Process (MDP), and aim to learn a robot manipulation policy $\pi$ that maps the state space $\mathcal{S}$ to the action space $\mathcal{A}$. The imitation dataset consists of $N$ demonstrations $\mathcal{D} = \{(s_0^i, a_0^i, s_1^i, a_1^i, ..., s_{H_i}^i)\}_{i=1}^{N}$ where each $s_0^i \sim D(\cdot)$ is sampled from the initial state distribution $D$. In this work, we use Behavioral Cloning [28] to train the policy with the objective $\arg\min_\theta \mathbb{E}_{(s,a)\sim\mathcal{D}}[-\log \pi_\theta(a|s)]$.

Figure 2: **MimicGen System Pipeline.** (left) MimicGen first parses the demos from the source dataset into segments, where each segment corresponds to an object-centric subtask (Sec. 4.1). (right) Then, to generate new demonstrations for a new scene, MimicGen generates and follows a sequence of end-effector target poses for each subtask by (1) choosing a segment from a source demonstration (chosen segments shown with blue border in figure above), (2) transforming it for the new scene, and (3) executing it (Sec. 4.2).

**Problem Statement and Assumptions.** Our goal is to use a **source dataset** $\mathcal{D}_{\mathrm{src}}$ that consists of a small set of human demonstrations collected on a task $\mathcal{M}$ and use it to generate a large dataset $\mathcal{D}$ on either the same task or **task variants** (where the initial state distribution $D$, the objects, or the robot arm can change). To generate a new demo: (1) a start state is sampled from the task we want to generate data for, (2) a demonstration $\tau \in \mathcal{D}_{\mathrm{src}}$ is chosen and adapted to produce a new robot trajectory $\tau'$, (3) the robot executes the trajectory $\tau'$ on the current scene, and if the task is completed successfully, the sequence of states and actions is added to the generated dataset $\mathcal{D}$ (see Sec. 4 for details of each step). We next outline some assumptions that our system leverages.

**Assumption 1: delta end effector pose action space.** The action space $\mathcal{A}$ consists of delta-pose commands for an end-effector controller and a gripper open/close command. This is a common action space used in prior work [3–7, 33]. This gives us an equivalence between delta-pose actions and controller target poses, and allows us to treat the actions in a demonstration as a sequence of target poses for the end effector controller (Appendix M).

**Assumption 2: tasks consist of a known sequence of object-centric subtasks.** Let $\mathcal{O} = \{o_1, ..., o_K\}$ be the set of objects in a task $\mathcal{M}$. As in Di Palo et al. [11], we assume that tasks consist of a sequence of object-centric subtasks $(S_1(o_{S_1}), S_2(o_{S_2}), ..., S_M(o_{S_M}))$, where the manipulation in each subtask $S_i(o_{S_i})$ is relative to a single object's coordinate frame ($o_{S_i} \in \mathcal{O}$). We assume this sequence is known (it is typically easy for a human to specify — see Appendix J).

**Assumption 3: object poses can be observed at the start of each subtask during data collection.** We assume that we can observe the pose of the relevant object $o_{S_i}$ at the start of each subtask $S_i(o_{S_i})$ during data collection (not, however, during policy deployment).

## 4  Method

We describe how MimicGen generates new demonstrations using a small source dataset of human demonstrations (see Fig. 2 for an overview). MimicGen first parses the source dataset into segments — one for each object-centric subtask in a task (Sec. 4.1). Then, to generate a demonstration for a new scene, MimicGen generates and executes a trajectory (sequence of end-effector control poses) for each subtask, by choosing a reference segment from the source demonstrations, transforming it according to the pose of the object in the new scene, and then executing the sequence of target poses using the end effector controller (Sec. 4.2).

### 4.1  Parsing the Source Dataset into Object-Centric Segments

Each task consists of a sequence of object-centric subtasks (Assumption 2, Sec. 3) — we would like to parse every trajectory $\tau$ in the source dataset into segments $\{\tau_i\}_{i=1}^{M}$, where each segment $\tau_i$ corresponds to a subtask $S_i(o_{S_i})$. In this work, to parse source demonstrations into segments for each subtask, we assume access to metrics that allow the end of each subtask to be detected automatically (see Appendix J for full details). After this step, every trajectory $\tau \in \mathcal{D}_{\mathrm{src}}$ has been split into a contiguous sequence of segments $\tau = (\tau_1, \tau_2, ..., \tau_M)$, one per subtask.

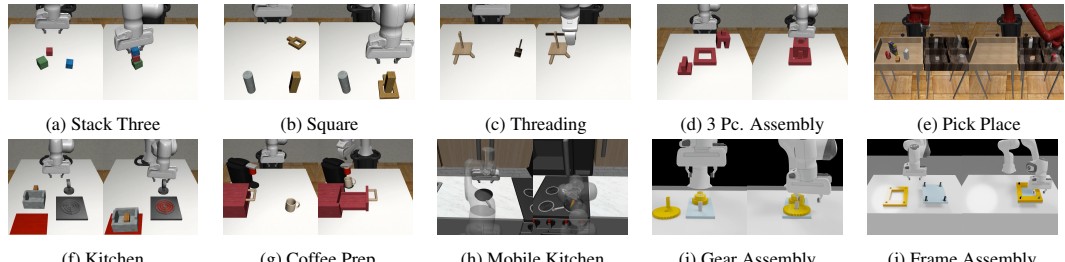

| (a) Stack Three | (b) Square | (c) Threading | (d) 3 Pc. Assembly | (e) Pick Place |
| (f) Kitchen | (g) Coffee Prep | (h) Mobile Kitchen | (i) Gear Assembly | (j) Frame Assembly |

Figure 3: **Tasks.** We use MimicGen to generate demonstrations for several tasks — these are a subset. They span a wide variety of behaviors including pick-and-place, insertion, interacting with articulated objects, and mobile manipulation, and include long-horizon tasks requiring chaining several behaviors together.

## 4.2 Transforming Source Data Segments for a New Scene

To generate a task demonstration for a new scene, MimicGen generates and executes a segment for each object-centric subtask in the task. As shown in Fig. 2 (right), this consists of three key steps for each subtask: (1) choosing a reference subtask segment in the source dataset, (2) transforming the subtask segment for the new context, and (3) executing the segment in the scene.

**Choosing a reference segment:** Recall that MimicGen parses the source dataset into segments that correspond to each subtask $\mathcal{D}_{\text{src}} = \{(\tau_1^j, \tau_2^j, ..., \tau_M^j)\}_{j=1}^N$ where $N = |\mathcal{D}_{\text{src}}|$. At the start of each subtask $S_i(o_{S_i})$, MimicGen chooses a corresponding segment from the set $\{\tau_i^j\}_{j=1}^N$. These segments can be chosen at random or by using the relevant object poses (more details in Appendix M).

**Transforming the source subtask segment:** We can consider the chosen source subtask segment $\tau_i$ for subtask $S_i(o_{S_i})$ as a sequence of target poses for the end effector controller (Assumption 1, Sec. 3). Let $T_B^A$ be the homogeneous 4×4 matrix that represents the pose of frame $A$ with respect to frame $B$. Then we can write $\tau_i = (T_W^{C_0}, T_W^{C_1}, ..., T_W^{C_K})$ where $C_t$ is the controller target pose frame at timestep $t$, $W$ is the world frame, and $K$ is the length of the segment. Since this motion is assumed to be relative to the pose of the object $o_{S_i}$ (frame $O_0$ with pose $T_W^{O_0}$) at the start of the segment, we will transform $\tau_i$ according to the new pose of the corresponding object in the current scene (frame $O_0'$ with pose $T_W^{O_0'}$) so that the relative poses between the target pose frame and the object frame are preserved at each timestep ($T_{O_0}^{C_t} = T_{O_0'}^{C_t'}$) resulting in the transformed sequence $\tau_i' = (T_W^{C_0'}, T_W^{C_1'}, ..., T_W^{C_K'})$ where $T_W^{C_t'} = T_W^{O_0'}(T_W^{O_0})^{-1}T_W^{C_t}$ (derivation in Appendix L). As an example, see how the source segment and transformed segment in the right side of Fig. 2 approach the mug in consistent ways. However, the first target pose of the new segment $T_W^{C_0'}$ might be far from the current end-effector pose of the robot in the new scene $T_W^{E_0'}$ (where $E$ is the end-effector frame). Consequently, MimicGen adds an **interpolation segment** at the start of $\tau_i'$ to interpolate linearly from the current end-effector pose ($T_W^{E_0'}$) to the start of the transformed segment $T_W^{C_0'}$.

**Executing the new segment:** Finally, MimicGen executes the new segment $\tau_i'$ by taking the target pose at each timestep, transforming it into a delta pose action (Assumption 1, Sec. 3), pairing it with the appropriate gripper open/close action from the source segment, and executing the new action.

The steps above repeat for each subtask until the final segment has been executed. However, this process can be imperfect — small trajectory deviations due to control and arm kinematics issues can result in task failure. Thus, MimicGen checks for task success after executing all segments, and only keeps successful demonstrations. We refer to the ratio between the number of successfully generated trajectories and the total number of attempts as the **data generation rate** (reported in Appendix O).

This pipeline only depends on object frames and robot controller frames — this enables data generation to take place across tasks with different initial state distributions, objects (assuming they have canonical frames defined), and robot arms (assuming they share a convention for the end effector control frame). In our experiments, we designed **task variants** for each robot manipulation task where we vary either the initial state distribution ($D$), an object in the task ($O$), or the robot arm ($R$), and showed that MimicGen enables data collection and imitation learning across these variants.

## 5 Experiment Setup

We applied MimicGen to a broad range of tasks (see Fig. 3) and task variants, in order to showcase how it can generate useful data for imitation learning across a diverse set of manipulation behaviors, including pick-and-place, contact-rich interactions, and articulation.

**Tasks and Task Variants.** Each task has a default reset distribution ($D_0$) (all source datasets were collected on this task variant), a broader reset distribution ($D_1$), and some have another ($D_2$), meant to pose even higher difficulty for data generation and policy learning. Consider the Threading task shown in Fig. 5 — in the $D_0$ variant, the tripod is always initialized in the same location, while in the $D_1$ variant, both the tripod and needle can move, and in the $D_2$ variant, the tripod and needle are randomized in novel regions of the workspace. In some experiments, we also applied MimicGen to task variants with a different robot arm ($R$) or different object instances ($O$) within a category.

We group the tasks into categories and summarize them below (full tasks and variants in Appendix K). Some tasks are implemented with the robosuite framework [49] (MuJoCo backend [50]) and others are implemented in Factory [51] (Isaac Gym [52] backend). **Basic Tasks** (Stack, Stack Three): a set of box stacking tasks. **Contact-Rich Tasks** (Square, Threading, Coffee, Three Piece Assembly, Hammer Cleanup, Mug Cleanup): a set of tasks that involve contact-rich behaviors such as insertion or drawer articulation. **Long-Horizon Tasks** (Kitchen, Nut Assembly, Pick Place, Coffee Preparation): require chaining multiple behaviors together. **Mobile Manipulation Tasks** (Mobile Kitchen): requires base and arm motion. **Factory Tasks** (Nut-Bolt-Assembly, Gear Assembly, Frame Assembly): a set of high-precision assembly tasks in Factory [51].

**Data Generation and Imitation Learning Methodology.** For each task, one human operator collected a source dataset of 10 demonstrations on the default variant ($D_0$) using a teleoperation system [2, 23] (with the exception of Mobile Kitchen, where we used 25 demos due to the large number of object variants, and Square, where we used 10 demos from the robomimic Square PH dataset [7]). MimicGen was used to generate 1000 demonstrations for each task variant, using each task's source dataset (full details in Appendix M). Since data generation is imperfect, each data generation attempt is not guaranteed to result in a task success. Attempts that did not achieve task success were discarded, and data collection kept proceeding for each task variant until 1000 task successes were collected. Each generated dataset was then used to train policies using Behavioral Cloning with an RNN policy [7]. We also adopt the convention from Mandlekar et al. [7] for reporting policy performance — the maximum success rate across all policy evaluations, across 3 different seeds (full training details in Appendix N). All policy learning results are shown on **image-based agents** trained with RGB observations (see Appendix P for low-dim agent results).

## 6 Experiments

We present experiments that (1) highlight the diverse array of situations that MimicGen can generate data for, (2) show that MimicGen compares favorably to collecting additional human demonstrations, both in terms of effort and downstream policy performance on the data, (3) offer insights into different aspects of the system, and (4) show that MimicGen can work on real-world robot arms.

### 6.1 Applications of MimicGen

We outline a number of applications that showcase useful properties of MimicGen.

**MimicGen data vastly improves agent performance on the source task.** A straightforward application of MimicGen is to collect a small dataset on some task of interest and then generate more data for that task. Comparing the performance of agents trained on the small source datasets vs. those trained on $D_0$ datasets generated by MimicGen, we see that there is substantial improvement across all our tasks (see Fig. 4). Some particularly compelling examples include Square (11.3% to 90.7%), Threading (19.3% to 98.0%), and Three Piece Assembly (1.3% to 82.0%).

**MimicGen data can produce performant agents across broad initial state distributions.** As shown in Fig. 4), agents trained using datasets generated on broad initial state distributions ($D_1$, $D_2$) are performant (42% to 99% on $D_1$), showing that MimicGen generates valuable datasets on new initial state distributions. In several cases, certain objects in the 10 source demonstrations never moved (the peg in Square, the tripod in Threading, the base in Three Piece Assembly, etc), but

| Task | Source | $D_0$ | $D_1$ | $D_2$ |
|------|--------|-------|-------|-------|
| Stack | $26.0 \pm 1.6$ | $100.0 \pm 0.0$ | $99.3 \pm 0.9$ | - |
| Stack Three | $0.7 \pm 0.9$ | $92.7 \pm 1.9$ | $86.7 \pm 3.4$ | - |
| Square | $11.3 \pm 0.9$ | $90.7 \pm 1.9$ | $73.3 \pm 3.4$ | $49.3 \pm 2.5$ |
| Threading | $19.3 \pm 3.4$ | $98.0 \pm 1.6$ | $60.7 \pm 2.5$ | $38.0 \pm 3.3$ |
| Coffee | $74.0 \pm 4.3$ | $100.0 \pm 0.0$ | $90.7 \pm 2.5$ | $77.3 \pm 0.9$ |
| Three Pc. Assembly | $1.3 \pm 0.9$ | $82.0 \pm 1.6$ | $62.7 \pm 2.5$ | $13.3 \pm 3.8$ |
| Hammer Cleanup | $59.3 \pm 5.7$ | $100.0 \pm 0.0$ | $62.7 \pm 4.7$ | - |
| Mug Cleanup | $12.7 \pm 2.5$ | $80.0 \pm 4.9$ | $64.0 \pm 3.3$ | - |
| Kitchen | $54.7 \pm 8.4$ | $100.0 \pm 0.0$ | $76.0 \pm 4.3$ | - |
| Nut Assembly | $0.0 \pm 0.0$ | $53.3 \pm 1.9$ | - | - |
| Pick Place | $0.0 \pm 0.0$ | $50.7 \pm 6.6$ | - | - |
| Coffee Preparation | $12.7 \pm 3.4$ | $97.3 \pm 0.9$ | $42.0 \pm 0.0$ | - |
| Mobile Kitchen | $2.0 \pm 0.0$ | $46.7 \pm 18.4$ | - | - |
| Nut-and-Bolt Assembly | $8.7 \pm 2.5$ | $92.7 \pm 2.5$ | $81.3 \pm 8.2$ | $72.7 \pm 4.1$ |
| Gear Assembly | $14.7 \pm 5.2$ | $98.7 \pm 1.9$ | $74.0 \pm 2.8$ | $56.7 \pm 1.9$ |
| Frame Assembly | $10.7 \pm 6.8$ | $82.0 \pm 4.3$ | $68.7 \pm 3.4$ | $36.7 \pm 2.5$ |

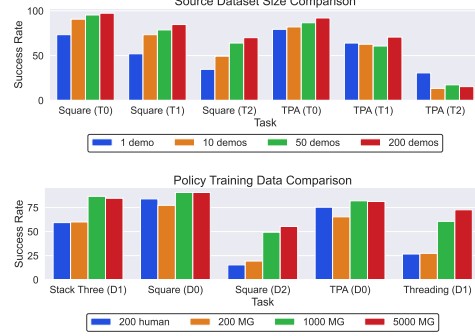

Figure 4: (left) **Agent Performance on Source and Generated Datasets.** Success rates (3 seeds) of image-based agents trained with BC on the 10 source demos and each 1000 demo MimicGen dataset. There is large improvement across all tasks on the default distribution ($D_0$) and agents are performant on the broader distributions ($D_1$, $D_2$). (top-right) **MimicGen with more source human demonstrations.** We found that using larger source datasets to generate MimicGen data did not result in significant agent improvement. (bottom-right) **Policy Training Dataset Comparison.** Image-based agent performance is comparable on 200 MimicGen demos and 200 human demos, despite MimicGen only using 10 source human demos. MimicGen can produce improved agents by generating larger datasets (200, 1000, 5000 demos), but there are diminishing returns.

data was generated (and policies consequently were trained) on regimes where the objects move in substantial regions of the robot workspace.

**MimicGen can generate data for different objects.** The source dataset in the Mug Cleanup task contains just one mug, but we generate demonstrations with MimicGen for an unseen mug ($O_1$) and for a set of 12 mugs ($O_2$). Policies trained on these datasets have substantial task success rates (90.7% and 75.3% respectively) (full results in Appendix F).

**MimicGen can generate data for diverse robot hardware.** We apply MimicGen to the Square and Threading source datasets (which use the Panda arm) and generate datasets for the Sawyer, IIWA, and UR5e across the $D_0$ and $D_1$ reset distribution variants. Interestingly, although the data generation rates differ greatly per arm (range 38%-74% for Square $D_0$), trained policy performance is remarkably similar across the 4 robot arms (80%-91%, full results in Appendix E). This shows the potential for using human demonstrations across robot hardware using MimicGen, an exciting prospect, as teleoperated demonstrations are typically constrained to a single robot.

**Applying MimicGen to mobile manipulation.** In the Mobile Kitchen task MimicGen yields a gain from 2.0% to 46.7% (image, Fig. 4) and 2.7% to 76.7% success rate (low-dim, Table P.1 in Appendix), highlighting that our method can be applied to tasks beyond static tabletop manipulation.

**MimicGen is simulator-agnostic.** We show that MimicGen is not limited to just one simulation framework by applying it to high-precision tasks (requiring **millimeter precision**) in Factory [51], a simulation framework built on top of Isaac Gym [52] to accurately simulate high-precision manipulation. We generate data for and train performant policies on the Nut-and-Bolt Assembly, Gear Assembly, and Frame Assembly tasks. Policies achieve excellent results on the nominal tasks ($D_0$) (82%-99%), a significant improvement over policies trained on the source datasets (9%-15%), and are also able to achieve substantial performance on wider reset distributions ($D_1$, $D_2$) (37%-81%).

**MimicGen can use demonstrations from inexperienced human operators and different tele-operation devices.** Surprisingly, policies trained on these MimicGen datasets have comparable performance to those in Fig. 4. See Appendix H for the full set of results.

## 6.2 Comparing MimicGen to using more human data

In this section, we contextualize the performance of agents trained on MimicGen data.

**Comparing task performance to prior works.** Zhu et al. [53] introduced the Hammer Cleanup and Kitchen tasks and reported agent performance on 100 human demonstrations for their method called BUDS. On Hammer Cleanup, BUDS achieved 68.6% ($D_0$), while BC-RNN achieves 59.3% on our 10 source demos, 100.0% on our generated 1000 $D_0$ demos, and 62.7% on the $D_1$ variant where both the hammer and drawer move substantially. On Kitchen, BUDS achieved 72.0% ($D_0$),

while BC-RNN achieves 54.7% on our 10 source demos, 100.0% on our generated $D_0$ data, and 76.0% on the $D_1$ variant, where all objects move in wider regions. This shows that using MimicGen to make effective use of a small number of human demonstrations can improve the complexity of tasks that can be learned with imitation learning. As another example, Mandlekar et al. [2] collected over 1000 human demos across 10 human operators on both the Nut Assembly and Pick Place tasks, but only managed to train proficient policies for easier, single-stage versions of these tasks using a combination of reinforcement learning and demonstrations. By contrast, in this work we are able to make effective use of just 10 human demonstrations to generate a set of 1000 demonstrations and learn proficient agents from them (76.0% and 58.7% low-dim, 53.3% and 50.7% image).

**Agent performance on data generated by MimicGen can be comparable to performance on an equal amount of human demonstrations.** We collect 200 human demonstrations on several tasks and compare agent performance on those demonstrations to agent performance on 200 demonstrations generated by MimicGen (see Fig. 4). In most cases, agent performance is similar, despite the 200 MimicGen demos being generated from just 10 human demos — a small number of human demos can be as effective (or even more) than a large number of them when used with MimicGen. MimicGen can also easily generate more demonstrations to improve performance (see Sec. 6.3), unlike the time-consuming nature of collecting more human data. This result also raises important questions on whether soliciting more human demonstrations can be redundant and not worth the labeling cost, and where to collect human demonstrations given a finite labeling bandwidth.

## 6.3 MimicGen Analysis

We analyze some practical aspects of the system, including (1) whether the number of source demonstrations used impacts agent performance, (2) whether the choice of source demonstrations matters, (3) whether agent performance can keep improving by generating more demonstrations, and (4) whether the data generation success rate and trained agent performance are correlated.

**Can dataset quality and agent performance be improved by using more source human demonstrations?** We used 10, 50, and 200 source human demonstrations on the Square and Three Piece Assembly tasks, and report the policy success rates in Fig. 4. We see that performance differences are modest (ranging from 2% to 21%). We also tried using just 1 human demo — in some cases performance was much worse (e.g. Square), while in others, there was no significant performance change (e.g. Three Piece Assembly). It is possible that performance could improve with more source human demos if they are curated in an intelligent manner, but this is left for future work.

**Does the choice of source human demonstrations matter?** For each generated dataset, we logged which episode came from which source human demonstration — in certain cases, this distribution can be very non-uniform. As an example, the generated Factory Gear Assembly task ($D_1$) had over 850 of the 1000 episodes come from just 3 source demonstrations. In the generated Threading task ($D_0$), one source demo had over 170 episodes while another had less than 10 episodes. In both cases, the number of attempted episodes per source demonstration was roughly uniform (since we picked them at random — details in Appendix M), but some were more likely to generate successful demonstrations than others. Furthermore, we found the source demonstration segment selection technique (Sec. 4.2) to matter for certain tasks (Appendix M). This indicates that both the initial set of source demos provided to MimicGen ($\mathcal{D}_{src}$), and how segments from these demos are chosen during each generation attempt ($\tau_i$ for each subtask, see Sec. 4.1) can matter.

**Can agent performance keep improving by generating more demonstrations?** In Fig. 4, we train agents on 200, 1000, and 5000 demos generated by MimicGen across several tasks. There is a large jump in performance from 200 to 1000, but not much from 1000 to 5000, showing that there can be diminishing returns on generating more data.

**Are the data generation success rate and trained agent performance correlated?** It is tempting to think that data generation success rate and trained agent performance are correlated, but we found that this is not necessarily true — there are datasets that had low dataset generation success rates (and consequently took a long time to generate 1000 successes) but had high agent performance after training on the data (Appendix O). A few examples are Object Cleanup ($D_0$) (29.5% generation rate, 82.0% agent rate), Three Piece Assembly ($D_0$) (35.6% generation rate, 74.7% agent rate), Coffee ($D_2$) (27.7% generation rate, 76.7% agent rate), and Factory Gear Assembly ($D_1$) (8.2% generation rate, 76.0% agent rate). These results showcase the value of using replay-based mechanisms for data collection instead of directly using them to deploy as policy as in prior works [8, 11].

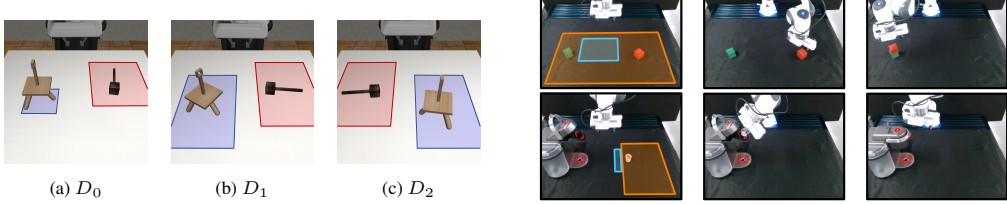

Figure 5: (left) **Reset Distributions.** Each task has a default reset distribution for the objects ($D_0$), a broader one ($D_1$), and some had a more challenging one ($D_2$). The figure shows the sampling regions for the tripod and needle in the Threading task. The tripod is at a fixed location in $D_0$, and $D_2$ swaps the relative locations of the tripod and needle. We generate data across diverse scene configurations by taking source demos from $D_0$ and generating data for all variants. (right) **Real Robot Tasks.** We apply MimicGen to two real robot tasks — Stack (top row) and Coffee (bottom row). In the first column, the blue and orange regions show the source ($D_0$) and generated ($D_1$) reset distributions for each task. We use 10 source demos per task, and generate 100 successful demos — MimicGen has a data generation success rate of 82.3% for Stack and 52.1% for Coffee.

## 6.4 Real Robot Evaluation

We validate that MimicGen can be applied to real-world robot arms and tasks. We collect 10 source demonstrations for each task in narrow regions of the workspace ($D_0$) and then generate demonstrations (200 for Stack, 100 for Coffee) for large regions of the workspace ($D_1$) (see Fig. 5). The generation success rate was 82.3% for Stack (243 attempts) and 52.1% for Coffee (192 attempts), showing that MimicGen works in the real world with a reasonably high success rate. We then trained visuomotor agents using a front-facing RealSense D415 camera and a wrist-mounted RealSense D435 camera (120×160 resolution). Over 50 evaluations, our Stack agent had 36% success rate and Coffee had 14% success rate (pod grasp success rate of 60% and pod insertion success rate of 20%). The lower numbers than from simulation might be due to the larger number of interpolation steps we used in the real world for hardware safety (50 total instead of 5) — these motions are difficult for the agent to imitate since there is little association between the intermediate motion and observations (**see Appendix G for more experiments and discussion**).

We also compared to agents trained on the source datasets (10 demos) in the narrow regions (orange regions in Fig. 5) where the source data came from — the Stack source agent had 0% success rate and the Coffee source agent had 0% success rate (with an insertion rate of 0% and pod grasp rate of 94%). The Coffee ($D_0$) task in particular has barely any variation (the pod can move vertically in a 5cm region) compared to the $D_1$ task, which is substantially harder (pod placed anywhere in the right half of the workspace). Agents trained with MimicGen data compare favorably to these agents, as they achieve non-zero success rates on broader task reset distributions.

## 7 Limitations

**See Appendix C for full set of limitations and discussion.** MimicGen assumes knowledge of the object-centric subtasks in a task and requires object pose estimates at the start of each subtask during data generation (Assumption 3, Sec. 3). MimicGen only filters data generation attempts based on task success, so generated datasets can be biased (Appendix Q). MimicGen uses linear interpolation between human segments (Appendix M.2), which does not guarantee collision-free motion, and can potentially hurt agent performance (Appendix G). MimicGen was demonstrated on quasi-static tasks with rigid objects, and novel objects were assumed to come from the same category.

## 8 Conclusion

We introduced MimicGen, a data generation system that can use small amounts of human demonstrations to generate large datasets across diverse scenes, object instances, and robots, and applied it to generate over 50K demos across 18 tasks from less than 200 human demos, including tasks involving long-horizon and high-precision manipulation. We showed that agents learning from this data can achieve strong performance. We further found that agent performance on MimicGen data can be comparable to performance on an equal number of human demos — this surprising result motivates further investigation into when to solicit additional human demonstrations instead of making more effective use of a small number, and whether human operator time would be better spent collecting data in new regions of the workspace. We hope that MimicGen motivates and enables exploring a more data-centric perspective on imitation learning in future work.

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

# Appendix

## A  Overview

We present several additional results in the Appendix.

- **FAQ** (Appendix B): answers to some common questions
- **Limitations** (Appendix C): more thorough list and discussion of MimicGen limitations
- **Full Related Work** (Appendix D): more thorough discussion on related work
- **Robot Transfer** (Appendix E): full set of results for generating data across robot arms
- **Object Transfer** (Appendix F): full set of results for generating data across objects
- **Real Robot Results** (Appendix G): additional details and discussion on the real robot experiments, including an explanation for the lower training results in the real world
- **Different Demonstrators** (Appendix H): results that show MimicGen works just as well when using source demos from suboptimal demonstrators and from different teleoperation devices
- **Motivation for MimicGen over Alternative Methods** (Appendix I): motivation for MimicGen over offline data augmentation and replay-based imitation
- **Additional Details on Object-Centric Subtasks** (Appendix J): more details and intuition on subtasks, including examples
- **Tasks and Task Variants** (Appendix K): detailed descriptions all tasks and task variants
- **Derivation of Subtask Segment Transform** (Appendix L): derivation of how MimicGen transforms subtask segments from the source data
- **Data Generation Details** (Appendix M): in-depth details on how MimicGen generates data
- **Policy Training Details** (Appendix N): details of how policies were trained from MimicGen datasets via imitation learning
- **Data Generation Success Rates** (Appendix O): data generation success rates for each of our generated datasets
- **Low-Dim Policy Training Results** (Appendix P): full results for agents trained on *low-dim* observation spaces (image agents presented in main text)
- **Bias and Artifacts in Generated Data** (Appendix Q): discussion on some undesirable properties of MimicGen data
- **Using More Varied Source Demonstrations** (Appendix R): investigation on whether having source demonstrations collected on a more varied set of task initializations is helpful
- **Data Generation with Multiple Seeds** (Appendix S): results that show there is very little variance in empirical results across different data generation seeds
- **Tolerance to Pose Estimation Error** (Appendix T): investigation of MimicGen's tolerance to pose error

## B  FAQ

1. **What are some limitations of MimicGen?**

   See Appendix C for a discussion.

2. **Why are policy learning results worse in the real world than in simulation?**

   See Appendix G for discussion and an additional experiment.

3. **Since data generation relies on open-loop replay of source human data, it seems like MimicGen only works for low-precision pick-and-place tasks.**

   We demonstrated that MimicGen can work for a large variety of manipulation tasks and behaviors beyond standard pick-and-place tasks. This includes tasks with non-trivial contact-rich manipulation (Gear Assembly has **1mm insertion tolerance**, and Picture Frame Assembly needs **alignment of 4 holes with 4mm tolerance each**), long-horizon manipulation (up to 8 subtasks), and behaviors beyond pick-and-place such as insertion, pushing, and articulation — see Appendix K for full details. The tasks also have pose variation well beyond typical prior works using BC from human demos [1, 3–7, 30, 33, 54, 55].

4. **Is MimicGen robust to noisy object pose estimates during data generation?**

   In the real world, we use the initial RGBD image to estimate object poses (see Appendix G). Thus, MimicGen is compatible with pose estimation methods and has some tolerance to pose error. We further investigated tolerance to pose estimate errors in simulation (see Appendix T) and found that while data generation rates can decrease (so data collection will take longer), policies trained on the generated data maintained the same level of performance.

5. **Several recent works apply offline data augmentation to existing datasets to create more data. What are the advantages of generating new data online like MimicGen does?**

   Offline data augmentation can be effective for generating larger dataset for robot manipulation [7, 35–45]; however, it can be difficult to generate plausible interactions without prior knowledge of physics [35] or causal dependencies [41, 42], especially for new scenes, objects, or robots. In contrast, by generating new datasets through environment interaction, MimicGen data is guaranteed to be physically-consistent. Additionally, in contrast to many offline data augmentation methods, MimicGen is easy to implement and apply in practice, since only a small number of assumptions are needed (see Sec. 3). See more discussion in Appendix I.2.

6. **What is the advantage of using replay-based imitation for data generation and then training a policy with BC (like MimicGen does) over using it as the final agent?**

   Replay-based imitation learning methods are promising for learning manipulation tasks using a handful of demonstrations [8–11, 46–48], but they have some limitations compared to MimicGen, which uses similar mechanisms during data generation, but trains an end-to-end closed-loop agent from the generated data. First, replay-based agents generally conform to a specific policy architecture, while MimicGen datasets allow full compatibility with a wide spectrum of offline policy learning algorithms [56]. Second, replay-based methods are typically *open-loop*, since they consist of replaying a demonstration blindly, while agents trained on MimicGen datasets can have *closed-loop*, reactive behavior, since the agent can respond to changes in observations. Finally, as we saw in Sec. 6 (and Appendix O), in many cases, the data generation success rate (a proxy for the performance of replay-based methods) can be significantly lower than the performance of trained agents. See more discussion in Appendix I.1.

7. **Why might a data generation attempt result in a failure?**

   One reason is that the interpolation segments are unaware of the geometry in the scene and consist of naive linear interpolation (see Appendix M.2), so these segments might result in unintended collisions. Another is that the way source segments are transformed do not consider arm kinematics, so the end effector poses where segments start might be difficult to reach. A third reason is that certain source dataset motions might be easier for the controller to track than others.

8. **When can MimicGen be applied to generate data for new objects?**

We demonstrated results on geometrically similar rigid-body objects from the same category (e.g. mugs, carrots, pans) with similar scales. We also assumed aligned canonical coordinate frames for all objects in a category, and that the objects are well-described by their poses (e.g. rigid bodies, not soft objects). Extending the system for soft objects or more geometrically diverse objects is left for future work.

9. **Can MimicGen data contain undesirable characteristics?**

   See Appendix Q for a discussion.

10. **Give a breakdown of how MimicGen was used to generate 50K demos from 200 human demos.**

    Here is the breakdown. It should be noted that this breakdown does not include our real robot demonstrations (200 demos generated from 20 source demos) or any extra datasets generated for additional experiments and analysis presented in the appendix.

    - 175 source demos: 10 source demos for each of 16 simulated tasks in Fig. 4 (except Mobile Kitchen, which has 25)
    - 36K generated demos: 1000 demos for each of the 36 task variants in Fig. 4
    - 12K generated demos: robot transfer experiment (Appendix E) had 2 tasks, each of which had 2 variants ($D_0$, $D_1$) and 3 new robot arms for $12 \times 1000$ demos.
    - 2K generated demos: object transfer experiment (Appendix F) had 1000 demos for the $O_1$ (new mug) and $O_2$ (12 mugs) variants.

# C Limitations

In this section, we discuss limitations of MimicGen that can motivate and inform future work.

1. **Known sequence of object-centric subtasks.** MimicGen assumes knowledge of the object-centric subtasks in a task (which object is involved at each subtask) and also assumes that this sequence of subtasks does not change (Assumption 2, Sec. 3).

2. **Known object poses at start of each subtask during data generation.** During data generation, at the start of each object-centric subtask, MimicGen requires an object pose estimate of the reference object for that subtask (Assumption 3, Sec 3). However, we demonstrated that we can run MimicGen in the real world, using pose estimation methods (Sec. 6.4 and Appendix G), and has some tolerance to errors in pose estimates (Appendix T). Another avenue for real world deployment is to generate data and train policies in simulation (where object poses are readily available) and then deploy simulation-trained agents in the real world [57–61] — this is left for future work.

3. **One reference object per subtask.** MimicGen assumes each task is composed of a sequence of subtasks that are each relative to exactly one object (Assumption 2, Sec. 3). Being able to support subtasks where the motion depends on more than one object (for example, placing an object relative to two objects, or on a cluttered shelf) is left for future work.

4. **Naive filtering for generated data.** MimicGen has a naive way to filter data generation attempts (just task success rates). However, this does not prevent the generated datasets from being biased, or having artifacts (see discussion in Appendix Q). Developing better filtering mechanisms is left for future work.

5. **Naive interpolation scheme and no guarantee on collision-free motion.** MimicGen uses a naive linear interpolation scheme to connect transformed human segments together (Appendix M.2). However, this method is not aware of scene geometry, and consequently can result in unintended collisions if objects happen to be in the way of the straight line path. We opted for this simple approach to avoid the complexity of integrating a planner and ensuring it uses the same action space (Operational Space Control [62]). We also saw that longer interpolation segments could be harmful to policy learning from generated data (Appendix G). Similarly, ensuring that motion plans are not harmful to policy learning could be non-trivial. Developing better-quality interpolation segments (e.g. potentially with motion planning) that are both amenable to downstream policy learning and safer for real-world operation is left for future work.

6. **Object transfer limitations.** While MimicGen can generate data for manipulating different objects (Appendix F), we only demonstrated results on geometrically similar rigid-body objects from the same category (e.g. mugs, carrots, pans) with similar scales. We also assumed aligned canonical coordinate frames for all objects in a category, and that the objects are well-described by their poses (e.g. rigid bodies, not soft objects). Extending the system for soft objects or more geometrically diverse objects is left for future work.

7. **Task limitations.** MimicGen was demonstrated on quasi-static tasks — it is unlikely to work on dynamic, non quasi-static tasks in its current form. However, a large number of robot learning works and benchmarks use quasi-static tasks [1, 3–7, 14, 18, 19, 22, 30, 33, 51, 54, 55, 63–65], making the system broadly applicable. We also did not apply MimicGen to tasks where objects had different dynamics from the source demonstrations (e.g. new friction values). However, there is potential for MimicGen to work, depending on the task. Recall that on each data generation attempt, MimicGen tracks a target end effector pose path (Sec. 4.2) — this allows data generation for robot arms with different dynamics (Appendix E), and could potentially allow it to work for different object dynamics (e.g. pushing a cube across different table frictions).

8. **Mobile manipulation limitations.** In Sec. 6.1, we presented results for MimicGen on the Mobile Kitchen task, which requires mobile manipulation (base and arm motion). Our current implementation has some limitations. First, it assumes that the robot does not move the mobile base and arm simultaneously. Second, we simply copy the mobile base actions from the reference segment rather than transforming it like we do for end effector actions. We found this simple approach sufficient for the Mobile Kitchen task (more details

in Appendix M.5). Future work could integrate more sophisticated logic for generating base motion (e.g. defining and using a reference frame for each base motion segment, like the object-centric subtasks used for arm actions, and/or integrating a motion planner for the base).

9. **No support for multi-arm tasks.** MimicGen only works for single arm tasks — extending it to generate datasets for multi-manual manipulation [25] is left for future work.

# D Full Related Work

This section presents a more thorough discussion of related work than the summary presented in the main text.

**Data Collection for Robot Learning.** There have been several data collection efforts to try and address the need for large-scale data in robotics. Some efforts have focused on self-supervised data collection where robots gather data on tasks such as grasping through trial-and-error [12–17]. RoboTurk [2, 23–26] is a system for crowdsourcing task demonstrations from human operators using smartphone-based teleoperation and video streams provided in web browsers. Several related efforts [3–6, 27] also collect large datasets (e.g. 1000s of demonstrations) by using a large number of human operators over extended periods of time. In contrast, MimicGen tries to make effective use of a small number of human demonstrations (e.g. 10) to generate large datasets. Some works have collected large datasets using pre-programmed demonstrators in simulation [18–22]; however, it can be difficult to scale these approaches up to more complex tasks, while we show that Mimic-Gen can be applied to a broad range of tasks. Prior work has also attempted to develop systems that can selectively query humans for demonstrations when they are needed, in order to reduce human operator time and burden [66–69]. In contrast, MimicGen only needs an operator to collect a few minutes of demonstrations at the start of the process. Generating large synthetic datasets has been a problem of great interest in other domains as well [70–76], and has also been used as a tool for benchmarking motion planning [77].

**Imitation Learning for Robot Manipulation.** Imitation Learning (IL) seeks to train policies from a set of demonstrations. Behavioral Cloning (BC) [28] is a standard method for learning policies offline, by training the policy to mimic the actions in the demonstrations. It has been used extensively in prior work for robot manipulation [1, 19, 25, 29–34] — in this work, we use BC to train single-task policies from datasets generated by MimicGen. However, MimicGen can also be used to generate datasets for a wide range of existing offline learning algorithms that learn from diverse multi-task datasets [53, 54, 78–82]. Some works have used offline data augmentation to increase the dataset size for learning policies [7, 35–45] — in this work we collect new datasets.

**Replay-Based Imitation Learning.** While BC is simple and effective, it typically requires several demonstrations to learn a task [7]. To alleviate this, many recent imitation learning methods try to learn policies from only a handful of demonstrations by *replaying* demonstrations in new scenes [8–11, 46–48]. Some methods [9–11] use trained networks that help the robot end effector approach poses from which a demonstration can be replayed successfully. In particular, Di Palo et al. [11] proposes an approach to replay parts of a single demonstration to solve multi-stage tasks — this is similar to the way MimicGen generates new datasets. However they make a number of assumptions that we do not (4D position and yaw action space vs. our 6-DoF action space, a single wrist camera view to enable spatial generalization). Furthermore, this work and others use demonstration replay as a component of the final trained agent — in contrast, we use it as a data generation mechanism. Consequently, these prior approaches are complementary to our data generation system, and in principle, could be used as a part of alternative schemes for data generation. In this work, we focus on the general framework of using such demonstration replay mechanisms to generate data that can be seamlessly integrated into existing imitation learning pipelines, and opt for an approach that emphasizes simplicity (more discussion in Appendix I). Our experiments also show that there can be a large benefit from collecting large datasets and training agents from them, instead of directly deploying a replay-based agent.

# E  Robot Transfer

In Sec. 6, we summarized results that show MimicGen can generate data for diverse robot hardware. Recall that we took the source datasets from the Square and Threading tasks (which use the Panda arm) and generated datasets for the Sawyer, IIWA, and UR5e robots across the $D_0$ and $D_1$ reset distribution variants (see Fig. E.1). Here, we present the complete set of results.

Notice that although the data generation rates have a large spread across robots (range 20%-74% for $D_0$, see Table E.1), the policy success rates are significantly higher and remarkably similar across robots (for example, 80%-91% on Square $D_0$ and 89%-98% on Threading $D_0$ — see the full image-based agent results in Table E.2 and low-dim agent results in Table E.3). This shows the potential for using human demonstrations across robot hardware using MimicGen, an exciting prospect, as teleoperated demonstrations are typically constrained to a single robot.

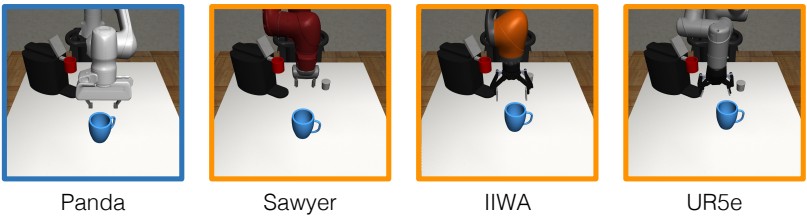

Figure E.1: **Robots used in Robot Transfer Experiment.** The figure shows the robot arms used for data generation. Source datasets were collected on the Panda arm (blue border) and used to generate data for the Sawyer, IIWA, and UR5e arms (orange border).

| Task Variant | Panda | Sawyer | IIWA | UR5e |
|---|---|---|---|---|
| Square ($D_0$) | 73.7 | 55.8 | 37.7 | 64.7 |
| Square ($D_1$) | 48.9 | 38.8 | 26.5 | 34.1 |
| Threading ($D_0$) | 51.0 | 28.8 | 20.4 | 21.4 |
| Threading ($D_1$) | 39.2 | 23.7 | 11.5 | 18.5 |

Table E.1: **Data Generation Rates on Different Robot Hardware.** The success rates of data generation are different across different robot arms (yet agents trained on these datasets achieve similar task success rates).

| Task Variant | Panda | Sawyer | IIWA | UR5e |
|---|---|---|---|---|
| Square ($D_0$) | $90.7 \pm 1.9$ | $86.0 \pm 1.6$ | $80.0 \pm 4.3$ | $84.7 \pm 0.9$ |
| Square ($D_1$) | $73.3 \pm 3.4$ | $60.7 \pm 2.5$ | $48.0 \pm 3.3$ | $56.0 \pm 4.3$ |
| Threading ($D_0$) | $98.0 \pm 1.6$ | $88.7 \pm 7.5$ | $94.0 \pm 3.3$ | $91.3 \pm 0.9$ |
| Threading ($D_1$) | $60.7 \pm 2.5$ | $50.7 \pm 3.8$ | $49.3 \pm 4.1$ | $60.7 \pm 2.5$ |

Table E.2: **Agent Performance on Different Robot Hardware.** We use MimicGen to produce datasets across different robot arms using the same set of 10 source demos (collected on the Panda arm) and train image-based agents on each dataset (3 seeds). The success rates are comparable across the different robot arms, indicating that MimicGen can generate high-quality data across robot hardware.

| Task Variant | Panda | Sawyer | IIWA | UR5e |
|---|---|---|---|---|
| Square ($D_0$) | $98.0 \pm 1.6$ | $87.3 \pm 1.9$ | $79.3 \pm 2.5$ | $82.0 \pm 1.6$ |
| Square ($D_1$) | $80.7 \pm 3.4$ | $69.3 \pm 2.5$ | $55.3 \pm 1.9$ | $67.3 \pm 3.4$ |
| Threading ($D_0$) | $97.3 \pm 0.9$ | $96.7 \pm 2.5$ | $93.3 \pm 0.9$ | $96.0 \pm 1.6$ |
| Threading ($D_1$) | $72.0 \pm 1.6$ | $73.3 \pm 2.5$ | $67.3 \pm 4.7$ | $80.0 \pm 4.9$ |

Table E.3: **Low-Dim Agent Performance on Different Robot Hardware.** We use MimicGen to produce datasets across different robot arms using the same set of 10 source demos (collected on the Panda arm) and train agents on each dataset (3 seeds). The success rates are comparable across the different robot arms, indicating that MimicGen can generate high-quality data across robot hardware.

# F Object Transfer

In Sec. 6, we summarized results that show MimicGen can generate data for different objects. Recall that we took the source dataset from the Mug Cleanup task and generated data with MimicGen for an unseen mug ($O_1$) and for a set of 12 mugs ($O_2$). Here, we present the complete set of results (Table F.1) and also visualize the mugs used for this experiment (Fig. F.1).

The Mobile Kitchen task that we generated data for also had different object variants — we show the 3 pans and 3 carrots in Fig. F.2. Results for this task are in Fig. 4 (image-based agents) and in Table P.1 (low-dim agents).

While these results are promising, we only demonstrated results on geometrically similar rigid-body objects from the same category (e.g. mugs, carrots, pans) with similar scales. We also assumed aligned canonical coordinate frames for all objects in a category, and that the objects are well-described by their poses (e.g. rigid bodies, not soft objects). Extending the system for soft objects or more geometrically diverse objects is left for future work.

| Task | $D_0$ | $O_1$ | $O_2$ |
|---|---|---|---|
| Mug Cleanup (DGR) | 29.5 | 31.0 | 24.5 |
| Mug Cleanup (SR, image) | $80.0 \pm 4.9$ | $90.7 \pm 1.9$ | $75.3 \pm 5.2$ |
| Mug Cleanup (SR, low-dim) | $82.0 \pm 2.8$ | $88.7 \pm 4.1$ | $66.7 \pm 2.5$ |

Table F.1: **Object Transfer Results.** We present data generation rates (DGR) and success rates (SR) of trained agents on the $O_1$ and $O_2$ variants of the Mug Cleanup task, which have an unseen mug, and a set of 12 mugs (a new mug per episode) respectively.

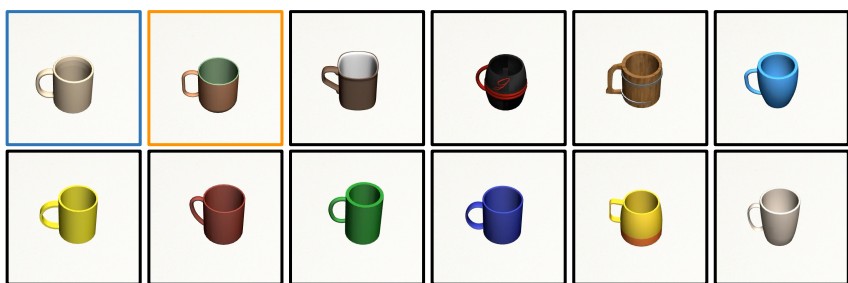

Figure F.1: **Objects used in Object Transfer Experiment.** The figure shows the mug used in the Mug Cleanup $D_0$ task (blue border), the unseen one in the $O_1$ task (orange border), and the complete set of mugs in the $O_2$ task.

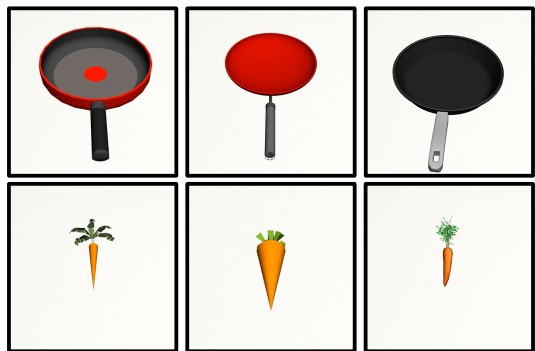

Figure F.2: **Objects used in Mobile Kitchen task.** The figure shows the 3 pans and 3 carrots used in the Mobile Kitchen task. On each episode a random pan and carrot are selected and initialized in the scene.

## G  Real Robot Results

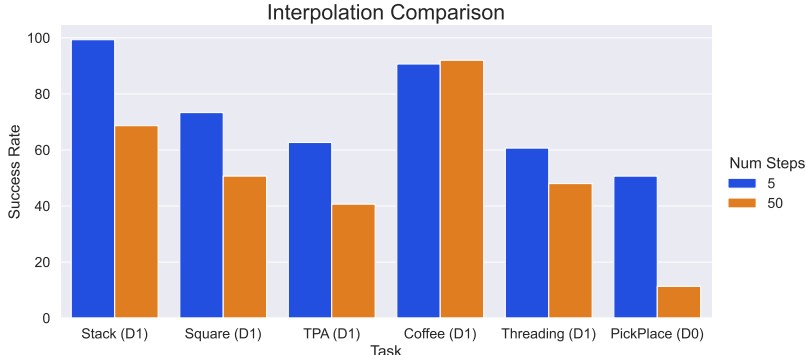

Figure G.1: **Effect of Increasing Interpolation Steps.** Comparing the effort of interpolation steps on trained image-based agents. Using an increased amount of interpolation can cause agent performance to decrease significantly. This could explain the gap between real-world and simulation agent performance.

In this section, we first provide further details on how we applied MimicGen to the real world tasks in Fig. 5, then we provide additional experiment results that help to explain the gap in trained policy performance between simulation and real.

**Real Robot Data Collection Details.** Recall that during data generation, MimicGen requires pose estimates at the start of each object-centric subtask (Assumption 3, Sec. 3). To do this, we use a front-view Intel RealSense D415 camera which has been calibrated (e.g. known extrinsics). We first convert the RGBD image to a point cloud and remove the table plane via RANSAC [83]. We then apply DBSCAN [84] clustering to identify object segments of interest, though alternative segmentation methods such as [85, 86] are also applicable. In the Stack task, the cube instances are distinguished by their color. In the Coffee task, the coffee machine and the pod are distinguished based on the segment dimensions. Finally for each identified object segment, we leverage [87] for global pose initialization, followed by ICP [88] refinement. Note that while the current pose estimation pipeline works reasonably well, our framework is not specific to certain types of perception methods. Recent [89–93] and future advances in state estimation could be used to apply MimicGen in real-world settings with less assumptions about the specific objects.

**Gap in Policy Performance between Sim and Real.** While we saw a significantly high data collection success rate (82.3% for Stack, 52.1% for Coffee), we saw much lower policy success rate on these tasks than in simulation (36% vs. 100% for Stack, and 14% vs. ~90% for Coffee), as described in Sec. 6). While there was considerably less data in the real world due to the time-consuming nature of real-world data collection (100 demos instead of 1000 demos), there were also other factors that could explain this gap.

As a safety consideration, our real-world tasks used much larger interpolation segments of $n_{interp} = 25$, $n_{fixed} = 25$ instead of the simulation default ($n_{interp} = 5$, $n_{fixed} = 0$) (see Appendix M.2 and Appendix M.6). We hypothesized that the increased duration of the interpolation segments made them difficult to imitate, since there was little association between the motion and what the agent sees in the observations (the motions are slow, and do not generally move towards regions of interest). To further investigate this, we ran an experiment in simulation where we used the same settings for interpolation for a subset of our tasks. The results are presented in Fig. G.1.

We see that for certain tasks, the larger interpolation segments cause agent performance to decrease significantly — for example image-based agents on Stack $D_1$ decrease from 99.3% success to 68.7% success, and image based agents on Pick Place decrease from 50.7% to 11.3%. These results confirm that the larger segments (together with the smaller dataset size) may have been responsible for lower real world performance. Developing better-quality interpolation segments that are both safe for real-world operation and amenable to downstream policy learning is left for future work.

Combining MimicGen with sim-to-real policy deployment methods [57–61, 94–97] is another exciting avenue for future work —simulation does not suffer from the same bottlenecks as real-world data collection (slow and time-consuming, requiring multiple arms and human supervisors to reset

the task), making simulation an ideal setting for MimicGen to generate large-scale diverse datasets. Recent sim2real efforts have been very promising — several works [60, 94–97] have been able to transfer policies trained via imitation learning from sim to real. Furthermore, MimicGen is entirely complementary to domain randomization techniques [98], which could also be applied to assist in transferring policies to the real world.

**Improved Performance with More Flexible Policy Models.** One promising avenue to improve real-world learning results is to develop and/or apply imitation learning algorithms that can better deal with multimodal and heterogeneous trajectories. We trained Diffusion Policy [99], a recent state-of-the-art imitation learning model, on our real-world Stack dataset. The new agent achieved a success rate of 76% across 50 evaluations – a significant improvement over the 36% success rate achieved by BC-RNN. This result provides an optimistic outlook on producing capable agents from real-world MimicGen data.

## H  Different Demonstrators

| Task | $D_0$ | $D_1$ | $D_2$ |
|---|---|---|---|
| Stack Three (Op. A, image) | $92.7 \pm 1.9$ | $86.7 \pm 3.4$ | - |
| Stack Three (Op. B, image) | $86.0 \pm 0.0$ | $69.3 \pm 5.0$ | - |
| Threading (Op. A, image) | $98.0 \pm 1.6$ | $60.7 \pm 2.5$ | $38.0 \pm 3.3$ |
| Threading (Op. B, image) | $98.0 \pm 1.6$ | $58.0 \pm 4.3$ | $38.0 \pm 8.6$ |
| Three Pc. Assembly (Op. A, image) | $82.0 \pm 1.6$ | $62.7 \pm 2.5$ | $13.3 \pm 3.8$ |
| Three Pc. Assembly (Op. B, image) | $76.0 \pm 1.6$ | $54.7 \pm 6.8$ | $5.3 \pm 1.9$ |
| Stack Three (Op. A, low-dim) | $88.0 \pm 1.6$ | $90.7 \pm 0.9$ | - |
| Stack Three (Op. B, low-dim) | $82.7 \pm 0.9$ | $84.0 \pm 3.3$ | - |
| Threading (Op. A, low-dim) | $97.3 \pm 0.9$ | $72.0 \pm 1.6$ | $60.7 \pm 6.2$ |
| Threading (Op. B, low-dim) | $97.3 \pm 0.9$ | $76.0 \pm 4.3$ | $70.0 \pm 1.6$ |
| Three Pc. Assembly (Op. A, low-dim) | $74.7 \pm 3.8$ | $61.3 \pm 1.9$ | $38.7 \pm 4.1$ |
| Three Pc. Assembly (Op. B, low-dim) | $77.3 \pm 2.5$ | $65.3 \pm 7.4$ | $46.0 \pm 9.1$ |

Table H.1: **MimicGen with Different Demonstrators.** We show that policies trained on MimicGen data can achieve similar performance even when the source demonstrations come from different demonstrators. Operator B used a different teleoperation device than Operator A, but policy training results on generated datasets are comparable for both image-based and low-dim agents.

| Task | $D_0$ | $D_1$ | $D_2$ |
|---|---|---|---|
| Square (Better, image) | $90.7 \pm 1.9$ | $73.3 \pm 3.4$ | $49.3 \pm 2.5$ |
| Square (Okay, image) | $90.0 \pm 1.6$ | $64.0 \pm 7.1$ | $50.0 \pm 2.8$ |
| Square (Worse, image) | $90.7 \pm 0.9$ | $59.3 \pm 2.5$ | $45.3 \pm 4.1$ |
| Square (Better, low-dim) | $98.0 \pm 1.6$ | $80.7 \pm 3.4$ | $58.7 \pm 1.9$ |
| Square (Okay, low-dim) | $95.3 \pm 0.9$ | $82.0 \pm 1.6$ | $60.7 \pm 1.9$ |
| Square (Worse, low-dim) | $95.3 \pm 0.9$ | $76.7 \pm 5.0$ | $52.7 \pm 1.9$ |

Table H.2: **MimicGen with Lower Quality Demonstrators.** We show that policies trained on MimicGen data can achieve similar performance even when the source demonstrations come from lower quality demonstrators. We compare across source datasets from the "Better", "Okay", and "Worse" subsets of the robomimic Square-MH dataset [7], which was collected by operators of different proficiency. Policy training results on generated datasets are comparable for both image-based and low-dim agents.

While most of our experiments use datasets from one particular operator, we show that Mimic-Gen can easily use demonstrations from different operators of mixed quality. We first collected 10 source demonstrations from a different operator on the Stack Three, Threading, and Three Piece Assembly tasks — this operator also used a different teleoperation device (3D mouse [49, 100]). We also used 10 demonstrations from one of the "Okay" operators and one of the "Worse" operators in the robomimic Square-MH dataset [7] to see if MimicGen could use lower-quality datasets. These source datasets were then provided to MimicGen to generate 1000 demonstrations for all task variants, and subsequently train policies — the results are summarized in Table H.1 (different demonstrator with different teleoperation device) and Table H.2 (lower quality demonstrators).

Interestingly, the operator using a different teleoperation interface produced policies that were extremely similar in performance to our original results (deviations of 0% to 17%). Furthermore, the policies produced from the datasets generated with the "Worse" and "Okay" operator data are also extremely similar in performance (deviations of 0% to 14%). This is quite surprising, as the robomimic study [7] found that there can be significant difficulty in learning from datasets produced by less experienced operators. **Our results suggest that in the large data regime, the harmful effects of low-quality data might be mitigated.** This is an interesting finding that can inform future work into learning from suboptimal human demonstrations [101–106].

# I  Motivation for MimicGen over Alternative Methods

In this section, we expand on the motivation for using data generation with MimicGen over two alternatives — replay-based imitation learning and offline data augmentation.

## I.1  Replay-Based Imitation Learning

Several recent works learn policies using only a handful of demonstrations by replaying the demonstrations in new scenes [8–11, 46–48]. While these methods are promising, there are some limitations. One limitation is that their learned policy usually uses demonstration replay as a part of their agent. This means that the policy is often composed of hybrid stages (such as a self-supervised network that learns to move the arm to configurations from which replay will be successful and a replay stage). By contrast, MimicGen uses a similar mechanism to *generate datasets* — this allows full compatibility with a wide spectrum of offline policy learning algorithms [56]. These datasets also allow for evaluating different design decisions (such as different observation spaces and learning methods), including the potential for multi-task benchmarks consisting of high-quality human data. Furthermore, by easily allowing datasets to be created and curated, MimicGen can facilitate future work to investigate how dataset composition can influence learned policy proficiency.

Another limitation is that replay-based imitation methods are typically *open-loop*, since they consist of replaying a demonstration blindly (the trajectory executed by the robot cannot adapt to small errors). By contrast, agents trained on MimicGen datasets can have *closed-loop*, reactive behavior, since the agent can respond to changes in observations.

Finally, as we saw in Sec. 6 (and Appendix O), in many cases, the data generation success rate (a proxy for the performance of replay-based methods) can be significantly lower than the performance of trained agents (one reason for this might be because of only training the policy on the successful data generation attempts, and another might be due to agent generalization).

## I.2  Offline Data Augmentation

Several works have used offline data augmentation to increase the dataset size for learning policies [7, 35–45]. Since this process is offline, it can greatly increase the size of the dataset. In fact, this can be complementary to MimicGen— we leverage pixel shift randomization [7, 36–39] when training image-based agents on MimicGen data.

However, because data augmentation is offline, it can be difficult to generate plausible interactions without prior knowledge of physics [35] or causal dependencies [41, 42], especially for new scenes, objects, or robots. Instead, MimicGen opts for generating new datasets through environment interaction by re-purposing existing human demonstrations — this automatically leads to physically-consistent data, since generation is online. In contrast to many offline data augmentation methods, MimicGen is easy to implement and apply in practice, since only a small number of assumptions are needed (see Sec. 3).

Similar to MimicGen, some recent works [43–45] have also shown an ability to create datasets with new objects, but these works typically change *distractor* objects that are not involved in manipulation — this leads to encouraging behavioral invariances (e.g. tell the policy to apply the same actions, even if the background and irrelevant objects are changed). By contrast, MimicGen generates datasets with new objects that are a critical part of the manipulation task — it seeks to generate data by adapting behavior to new contexts.

# J  Additional Details on Object-Centric Subtasks

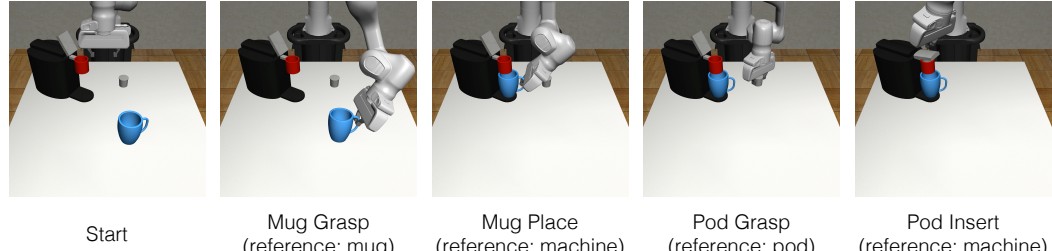

Figure J.1: **Illustrative Example of Object-Centric Subtasks.** In this example, the robot must prepare a cup of coffee by placing the mug on the machine, and the coffee pod into the machine. This task is easily broken down into a sequence of object-centric subtasks — this figure shows the end of each subtask, and the relevant object for each subtask. There is a mug grasping subtask (motion relative to mug), a mug placement subtask (motion relative to machine), a pod grasping subtask (motion relative to pod), and a pod insertion subtask (motion relative to machine). The robot can solve this task by sequencing motions relative to each object frame (one per subtask).

Object-centric subtasks (Assumption 2 in Sec. 3) are a key part of how MimicGen generates new demonstrations. In this section, we provide more details on how they are defined, and how subtask segments are parsed from the source demonstrations. We also show some examples to build intuition.

## J.1  How Tasks can be broken up into Object-Centric Subtasks

We first restate Assumption 2 — we assume that **tasks consist of a known sequence of object-centric subtasks**. Let $\mathcal{O} = \{o_1, ..., o_K\}$ be the set of objects in a task $\mathcal{M}$. As in Di Palo et al. [11], we assume that tasks consist of a sequence of object-centric subtasks $(S_1(o_{S_1}), S_2(o_{S_2}), ..., S_M(o_{S_M}))$, where the manipulation in each subtask $S_i(o_{S_i})$ is relative to a single object's ($o_{S_i} \in \mathcal{O}$) coordinate frame. We assume this sequence is known.

Specifying the sequence of object-centric subtasks is generally easy and intuitive for a human to do. As a first example, consider the coffee preparation task shown in Fig. J.1 (and Fig. 2). A robot must prepare a cup of coffee by grasping a mug, placing it on the coffee machine, grasping a coffee pod, inserting the pod into the machine, and closing the machine lid. This task can be broken down into a sequence of object-centric subtasks: a mug-grasping subtask (motion is relative to mug), a mug-placement subtask (motion relative to machine), a pod-grasping subtask (motion relative to pod), and a final pod-insertion and lid-closing subtask (motion relative to machine). Consequently, the robot can solve this task by sequencing several object-centric motions together. This is the key idea behind how MimicGen data generation works — it takes a set of source human demos, breaks them up into segments (where each segment solves a subtask), and then applies each subtask segment in a new scene. The subtasks are visualized in Fig. J.1.

We also emphasize that a wide variety of tasks can be broken down into object-centric subtasks (e.g. Assumption 2 applies to a wide variety of tasks, especially those that are commonly considered in the robot learning community). Fig. J.2 illustrates subtasks for some of our tasks (more discussion in Appendix J.3 below).

## J.2  Parsing the Source Dataset into Object-Centric Subtask Segments

We now provide more details on the parsing procedure described in Sec. 4.1. Recall that we would like to parse every trajectory $\tau$ in the source dataset into segments $\{\tau_i\}_{i=1}^M$, where each segment $\tau_i$ corresponds to a subtask $S_i(o_{S_i})$. We assume access to metrics that allow the end of each subtask to be detected automatically. In our running example from Fig. 2, this would correspond to metrics that use the state of the robot and objects to detect when the mug grasp, mug placement, pod grasp, and machine lid close occurs. This information is usually readily available in simulation, as it is often required for checking task success. With these metrics, we can easily run through the set of demonstrations, detect the end of each subtask sequentially, and use those as the subtask

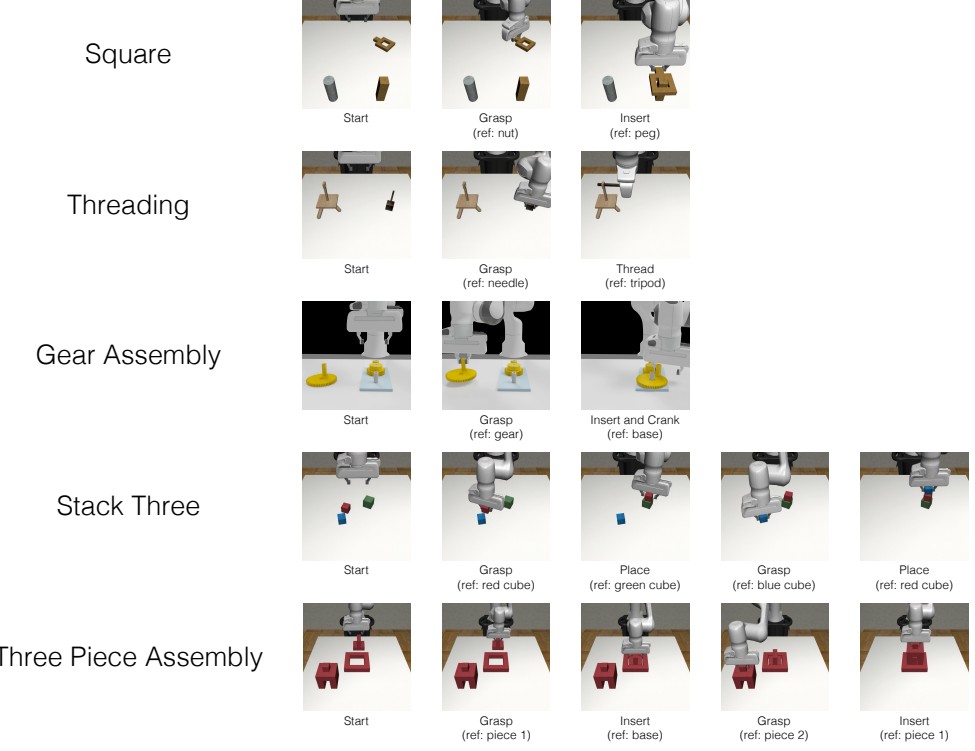

Figure J.2: **Object-Centric Subtasks for Selected Tasks** This figure shows the end of each object-centric subtask (and the reference object) for a subset of the tasks in the main text. MimicGen assumes that this subtask structure is known for each task; however, specifying this subtask structure is generally easy and intuitive for a human.

boundaries, to end up with every trajectory $\tau \in \mathcal{D}_{\text{src}}$ split into a contiguous sequence of segments $\tau = (\tau_1, \tau_2, ..., \tau_M)$, one per subtask.

**However, another alternative that requires no privileged information (and hence is suitable for real world settings) is to have a human manually annotate the end of each subtask.** As the number of source demonstrations is usually small, this is easy for a human operator to do, either while collecting each demonstration or annotating them afterwards. In this work, we opted for the former method (automated subtask end metrics) because they were readily available for our tasks or easy to craft.

### J.3 Specific Examples

We provide some examples in this section of how some tasks are broken up into object-centric subtasks. The examples are provided in Fig. J.2. For each task below, we outline the object-centric subtasks, and the subtask end detection metrics used for parsing the source human demos into segments that correspond to each subtask. Note that these metrics are only used for parsing the source human demos and are not assumed to be available during policy execution.

**Square.** There are 2 subtasks — grasping the nut (motion relative to nut) and inserting the nut onto the peg (motion relative to peg). To detect the end of the grasp subtask, we check for contact between the robot fingers and the nut. For the insertion subtask, we just use the task success check.

**Threading.** There are 2 subtasks — grasping the needle (motion relative to needle) and threading the needle into the tripod (motion relative to tripod). To detect the end of the grasp subtask, we check for contact between the robot fingers and the needle. For the threading subtask, we just use the task success check.

**Gear Assembly.** There are 2 subtasks — grasping the gear (motion relative to gear) and inserting the gear into the base and turning the crank (motion relative to base). To detect the end of the grasp subtask, we check if the gear has been lifted by a threshold. For the insertion subtask, we just use the task success check.

**Stack Three.** There are 4 subtasks — grasping the red block (motion relative to red block), placing the red block onto the green block (motion relative to green block), grasping the blue block (motion relative to blue block), and placing the blue block onto the red block (motion relative to red block). To detect the end of each grasp subtask we check for contact between the robot fingers and the relevant block. For each place subtask, we check that the relevant block has been lifted and is in contact with the block that should be underneath it.

**Three Piece Assembly.** There are 4 subtasks — grasping the first piece (motion relative to first piece), inserting the first piece into the base (motion relative to base), grasping the second piece (motion relative to second piece), and inserting the second piece onto the first piece (motion relative to first piece). To detect the end of each grasp subtask, we check for contact between the robot fingers and the relevant piece. For each insertion subtask, we re-use the insertion check from the task success check.

# K Tasks and Task Variants

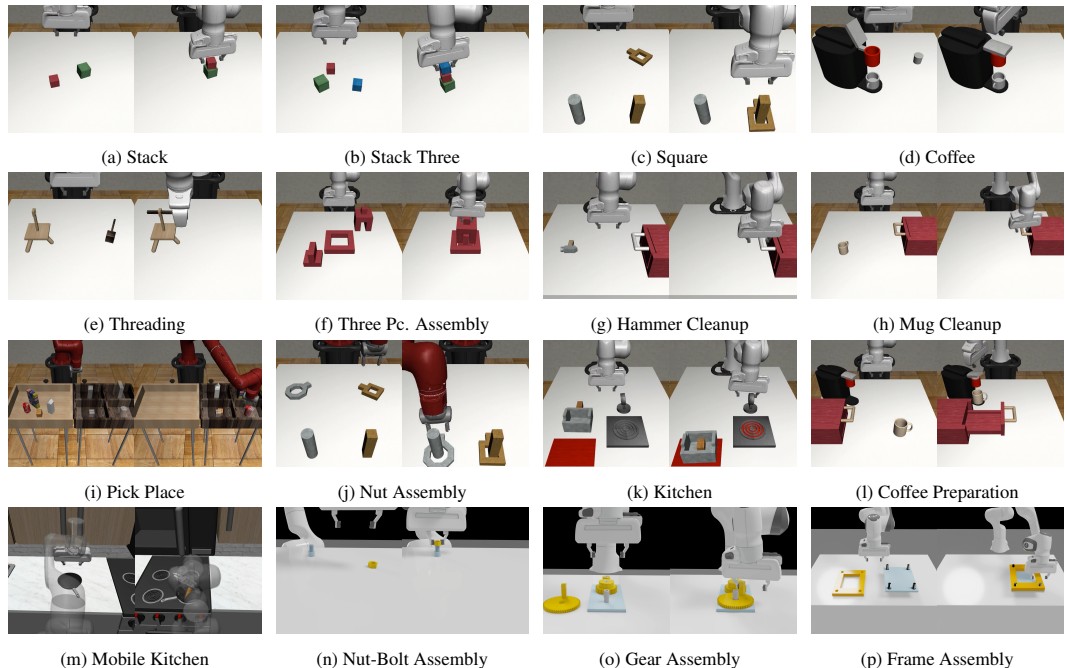

|  |  |  |  |
|---|---|---|---|
| (a) Stack | (b) Stack Three | (c) Square | (d) Coffee |
| (e) Threading | (f) Three Pc. Assembly | (g) Hammer Cleanup | (h) Mug Cleanup |
| (i) Pick Place | (j) Nut Assembly | (k) Kitchen | (l) Coffee Preparation |
| (m) Mobile Kitchen | (n) Nut-Bolt Assembly | (o) Gear Assembly | (p) Frame Assembly |

Figure K.1: **Tasks (all).** We show all of the simulation tasks in the figure above. They span a wide variety of behaviors including pick-and-place, precise insertion and articulation, and mobile manipulation, and include long-horizon tasks requiring chaining several behaviors together.

In this section, we provide more detailed descriptions of each of our tasks and task variants. The tasks (Fig. K.1) and task variants (especially their reset distributions) are best appreciated on the website (https://sites.google.com/view/corl2023mimicgen/home). We group the tasks into categories as in Sec. 5 and describe the goal, the variants, and the object-centric subtasks in each task. As mentioned in Sec. 3 and Appendix. M.1, the tasks have a delta-pose action space (implemented with an Operational Space Controller [62]). Control happens at 20 hz.

**Basic.** A basic set of box stacking tasks.

- **Stack [49]** Stack a red block on a green one. Blocks are initialized in a small (0.16m x 0.16m) region ($D_0$) and a large (0.4m x 0.4m) region ($D_1$) with a random top-down rotation. There are 2 subtasks (grasp red block, place onto green). We also develop a version of this task in the real-world (Fig. 5) , where the $D_0$ region is a 0.21m x 0.30m box and the $D_1$ region is a 0.44m x 0.85m box.

- **Stack Three.** Same as Stack, but additionally stack a blue block on the red one. Blocks are initialized in a small (0.20m x 0.20m) region ($D_0$) and a large (0.4m x 0.4m) region ($D_1$) with a random top-down rotation. There are 4 subtasks (grasp red block, place onto green, grasp blue block, place onto red).

**Contact-Rich.** A set of tasks that involve contact-rich behaviors such as insertion or drawer articulation. In each $D_0$ variant, at least one object never moves.

- **Square [7].** Pick a square nut and place on a peg. ($D_0$) Peg never moves, nut is placed in small (0.005m x 0.115m) region with a random top-down rotation. ($D_1$) Peg and nut move in large regions, but peg rotation fixed. Peg is initialized in 0.4m x 0.4m box and nut is initialized in 0.23m x 0.51m box. ($D_2$) Peg and nut move in larger regions (0.5m x 0.5m box of initialization for both) and peg rotation also varies. There are 2 subtasks (grasp nut, place onto peg).

- **Threading [24].** Pick a needle and thread through a hole on a tripod. ($D_0$) Tripod is fixed, needle moves in modest region (0.15m x 0.1m box with 60 degrees of top-down rotation variation). ($D_1$) Tripod and needle move in large regions on the left and right portions of the table respectively. The needle is initialized in a 0.25m x 0.1m box with 240 degrees

of top-down rotation variation and the tripod is initialized in a 0.25m x 0.1m box with 120 degrees of top-down rotation variation. $(D_2)$ Tripod and needle are initialized on the right and left respectively (reversed from $D_1$). The size of the regions is the same as $D_1$. There are 2 subtasks (grasp needle, thread into tripod).

- **Coffee [24].** Pick a coffee pod, insert into coffee machine, and close the machine hinge. $(D_0)$ Machine never moves, pod moves in small (0.06m x 0.06m) box. $(D_1)$ Machine and pod move in large regions on the left and right portions of the table respectively. The machine is initialized in a 0.1m x 0.1m box with 90 degrees of top-down rotation variation and the pod is initialized in a 0.25m x 0.13m box. $(D_2)$ Machine and pod are initialized on the right and left respectively (reversed from $D_1$). The size of the regions is the same as $D_1$. We also develop a version of this task in the real-world (Fig. 5) – in $D_0$, the pod is initialized in a 0.05m vertical strip and in $D_1$, the pod is initialized in a 0.44m x 0.35m box. There are 2 subtasks (grasp pod, insert-into and close machine).

- **Three Piece Assembly.** Pick one piece, insert it into the base, then pick the second piece, and insert into the first piece to assemble a structure. $(D_0)$ base never moves, both pieces move around base with fixed rotation in a 0.44m x 0.44m region. $(D_1)$ All three pieces move in workspace (0.44m x 0.44m region) with fixed rotation. $(D_2)$ All three pieces can rotate (the base has 90 degrees of top-down rotation variation, and the two pieces have 180 degrees of top-down rotation variation). There are 4 subtasks (grasp piece 1, place into base, grasp piece 2, place into piece 2).

- **Hammer Cleanup [53].** Open drawer, pick hammer, and place into drawer, and close drawer. $(D_0)$ Drawer is fixed, and hammer initialized in a small 0.08m x 0.07m box with 11 degrees of top-down rotation variation. $(D_1)$ Drawer and hammer both move in large regions. The drawer is initialized in a 0.2m x 0.1m box with 60 degrees of top-down rotation variation and the hammer is initialized in a 0.4m x 0.12m box with a random top-down rotation. There are 3 subtasks (open drawer, grasp hammer, place into drawer and close).

- **Mug Cleanup.** Similar to Hammer Cleanup but with a mug and with additional variants. $(D_0)$ The drawer does not move and the mug moves in a 0.3m x 0.15m box with a random top-down rotation. $(D_1)$ The mug moves in a 0.2m x 0.1m box with 60 degrees of top-down rotation variation and the mug is initialized in a 0.4m x 0.15m box with a random top-down rotation. $(O_1)$ A different mug is used. $(O_2)$ On each task reset, one of 12 mugs is sampled. There are 3 subtasks as in Hammer Cleanup.

**Long-Horizon.** A set of tasks that require chaining multiple behaviors together.

- **Kitchen [53].** Switch stove on, place pot onto stove, place bread into pot, place pot in front of serving region and push it there, and turn off the stove. $(D_0)$ The bread is initialized in a 0.03m x 0.06m region with fixed rotation and the pot is initialized in a 0.005m x 0.02m region with 11 degrees of top-down rotation variation. The other items do not move. $(D_1)$ Bread, pot, stove, button, and serving region all move in wider regions. Bread: 0.2m x 0.2m box with 180 degree top-down rotation variation, pot: 0.1m x 0.15m box with 60 degrees top-down rotation variation, stove: 0.17m x 0.1505m box with fixed rotation, button: 0.26m x 0.15m box with fixed rotation, serving region: 0.15m horizontal strip. There are 7 subtasks (turn stove on, grasp pot, place pot on stove, grasp bread, place bread in pot, serve pot onto serving region, and turn stove off).

- **Nut Assembly [49].** Similar to Square, but place both a square nut and round nut onto two different pegs. $(D_0)$ Each nut is initialized in a small box (0.005m x 0.115m region with a random top-down rotation). There are 4 subtasks (grasp each nut and place onto each peg).

- **Pick Place [49].** Place four objects into four different bins. $(D_0)$ Objects are initialized anywhere within the large box (0.29m x 0.39m). We use a slightly simpler version of this task where the objects are initialized with top-down rotations between 0 and 90 degrees (instead of any top-down rotation). There are 8 subtasks (grasp each obejct and place into each bin).

- **Coffee Preparation.** A full version of Coffee — load mug onto machine, open machine, retrieve coffee pod from drawer and insert into machine. $(T_0)$ The mug moves in modest (0.15m x 0.15m) region with fixed top-down rotation and the pod inside the drawer moves

in a 0.06m x 0.08m region while the machine and drawer are fixed. ($T_1$) The mug is initialized in a larger region (0.35m x 0.2m box with uniform top-down rotation) and the machine also moves in a modest region (0.1m x 0.05m box with 60 degrees of top-down rotation variation). There are 5 subtasks (grasp mug, place onto machine and open lid, open drawer, grasp pod, insert into machine and close lid).

**Mobile Manipulation.** Tasks involving mobile manipulation.

- **Mobile Kitchen.** Set up frying pan, by retrieving a pan from counter and placing onto stove, followed by retrieving a carrot from sink and placing onto pan. ($D_0$) The pan starts in a 0.2m x 0.4m region in the center of the countertop (with 120 degrees of top-down rotation variation) and the carrot starts in a 0.1m x 0.1m region inside the sink (with 60 degrees of rotation variation). There are three possible pans and three possible carrots sampled randomly for each episode. There are 4 subtasks (grasp gap, place pan, grasp carrot, place carrot). The latter three stages involve operating the mobile base.

**Factory.** A set of high-precision tasks in Factory [51].

- **Nut-and-Bolt Assembly.** Pick nut and align onto a bolt. ($D_0$) Nut and bolt are initialized in modest regions of size 0.2m x 0.2m with no rotation variation. ($D_1$) Nut and bolt initialized anywhere in workspace (0.35m x 0.8m box) with fixed rotation. ($D_2$) Nut and bolt can rotate (180 degrees of top-down rotation variation). There are 2 subtasks (pick nut and place onto bolt)

- **Gear Assembly.** Pick a gear, insert it onto a shaft containing other gears, and turn the gear crank to move the other gears. ($D_0$) Base is fixed, and gear moves in modest region (0.1m x 0.1m with no rotation variation). ($D_1$) Base and gear move in larger regions (of size 0.3m x 0.3m) with fixed rotation. ($D_2$) Both move with rotations (180 degrees of top-down variation for the gear and 90 degrees of top-down variation for the base). There are 2 subtasks (grasp gear, insert into base and crank).

- **Frame Assembly.** Pick a picture frame border with 4 holes and insert onto a base with 4 bolts rigidly attached. ($D_0$) Frame border and base move in small regions of size 0.1m x 0.1m with fixed rotation. ($D_1$) Frame border and base move in much larger regions of size 0.3m x 0.3m with fixed rotation. ($D_2$) Both move with rotations (60 degrees of top-down variation for both). There are 2 subtasks (grasp frame border and insert into base).

## L  Derivation of Subtask Segment Transform

In this section, we provide a complete derivation of the source subtask segment transformation presented in Sec. 4.2. Recall that $T_B^A$ denotes a homogenous $4{\times}4$ matrix that represents the pose of frame $A$ with respect to frame $B$. We have chosen a source subtask segment consisting of target poses for the end effector controller (Assumption 1, Sec. 3) $\tau_i = (T_W^{C_0}, T_W^{C_1}, ..., T_W^{C_K})$ where $C_t$ is the controller target pose frame at timestep $t$, $W$ is the world frame, and $K$ is the length of the segment.

We would like to transform $\tau_i$ according to the new pose of the corresponding object in the current scene (frame $O_0'$ with pose $T_W^{O_0'}$) so that the relative poses between the target pose frame and the object frame are preserved at each timestep ($T_{O_0'}^{C_t'} = T_{O_0}^{C_t}$). We can write $T_{O_0'}^{C_t'} = (T_W^{O_0'})^{-1} T_W^{C_t'}$ and $T_{O_0}^{C_t} = (T_W^{O_0})^{-1} T_W^{C_t}$. Setting them equal, we have

$$(T_W^{O_0'})^{-1} T_W^{C_t'} = (T_W^{O_0})^{-1} T_W^{C_t}$$

Rearranging for $T_W^{C_t'}$ by left-multiplying by $T_W^{O_0'}$ we obtain

$$T_W^{C_t'} = T_W^{O_0} (T_W^{O_0'})^{-1} T_W^{C_t}$$

which is the equation we use to transform the source segment.

## M   Data Generation Details

In this section, we provide additional details on how MimicGen generates data. We first provide additional details about components of MimicGen that were not discussed in the main text. This includes further discussion on how MimicGen converts between delta-pose actions and controller target poses (Appendix M.1), more details on how interpolation segments are generated (Appendix M.2), an overview of different ways the reference segment can be selected (Appendix M.3), details on how transformed trajectories are executed with action noise (Appendix M.4), additional details on our pipeline for mobile manipulation tasks (Appendix M.5), and finally, a list of the data generation hyperparameters for each task (Appendix M.6).

### M.1   Equivalence between delta-pose actions and controller target poses

We assume that the action space $\mathcal{A}$ consists of delta-pose commands for an end effector controller (Assumption 1, Sec. 3). As in [7], we assume that actions are 7-dimensional, where the first 3 components are the desired translation from the current end effector position, the next 3 components represent the desired delta rotation from the current end effector rotation, and the final component is the gripper open/close action. The delta rotation is represented in axis-angle form, where the magnitude of the 3-vector gives the angle, and the unit vector gives the axis. The robot controller converts the delta-pose action into an absolute pose target $T_W^C$ by adding the delta translation to the current end effector position, and applying the delta rotation to the current end effector rotation.

Consequently, at each timestep in a demonstration $\{s_t, a_t\}_{t=1}^T$, it is possible to convert each action $a_t$ to a controller pose target $T_W^{C_t}$ by using the end effector pose at each timestep. MimicGen uses this to represent each segment in the source demonstration as a sequence of controller poses. MimicGen also uses this conversion to execute a new transformed segment during data generation — it converts the sequence of controller poses in the segment to a delta-pose action at each timestep during execution, using the current end effector position.

### M.2   Details on Interpolation Segments

As mentioned in Sec. 4.2, MimicGen adds an interpolation segment at the start of each transformed segment during data generation to interpolate from the current end effector pose $T_W^{E_0'}$ and the start of the transformed segment $T_W^{C_0'}$. There are two relevant hyperparameters for the interpolation segment in each subtask segment — $n_{\text{interp}}$ and $n_{\text{fixed}}$. We first use simple linear interpolation between the two poses (linear in position, and spherical linear interpolation for rotation) to add $n_{\text{interp}}$ intermediate controller poses between $T_W^{E_0'}$ and $T_W^{C_0'}$, and then we hold $T_W^{C_0'}$ fixed for $n_{\text{fixed}}$ steps. These intermediate poses are all added to the start of the transformed segment, and given to MimicGen to execute one by one.

### M.3   Reference Segment Selection

Recall that MimicGen parses the source dataset into segments that correspond to each subtask $\mathcal{D}_{\text{src}} = \{(\tau_1^j, \tau_2^j, ..., \tau_M^j)\}_{j=1}^N$ (Sec. 4.1). During data generation, at the start of each subtask $S_i(o_{S_i})$, MimicGen must choose a corresponding segment from the set $\{\tau_i^j\}_{j=1}^N$ of $N$ subtask segments in $\mathcal{D}_{\text{src}}$. It suffices to choose only one source demonstration $j \in \{1, 2, ..., N\}$ since this uniquely identifies the subtask segment for the current subtask. We discuss some variants of how this selection occurs.

**Selection Frequency.** As presented in the main text (Fig. 2), MimicGen can select a source demonstration $j$ (and corresponding segment) at the start of each subtask. However, in many cases, this can be undesirable, since different demonstrations might have used different strategies that are incompatible with each other. As an example, two demonstrations might have different object grasps for the mug in Fig. 2 — each grasp might require a different placement strategy. Consequently, we introduce a hyperparameter, **per-subtask**, which can toggle this behavior — if it is set to False, MimicGen chooses a single source demonstration $j$ at the start of a data generation episode and holds it fixed (so all source subtask segments are from the same demonstration, $(\tau_1^j, \tau_2^j, ..., \tau_M^j)$).

| Task | normal | no noise | replay w/ noise |
|---|---|---|---|
| Square ($D_0$) (DGR) | 73.7 | 80.5 | 88.1 |
| Square ($D_1$) (DGR) | 48.9 | 50.7 | - |
| Square ($D_2$) (DGR) | 31.8 | 33.4 | - |
| Threading ($D_0$) (DGR) | 51.0 | 84.5 | 53.8 |
| Threading ($D_1$) (DGR) | 39.2 | 50.8 | - |
| Threading ($D_2$) (DGR) | 21.6 | 27.3 | - |
| Square ($D_0$) (SR, image) | $90.7 \pm 1.9$ | $72.0 \pm 3.3$ | $42.0 \pm 1.6$ |
| Square ($D_1$) (SR, image) | $73.3 \pm 3.4$ | $56.7 \pm 0.9$ | - |
| Square ($D_2$) (SR, image) | $49.3 \pm 2.5$ | $42.7 \pm 6.6$ | - |
| Threading ($D_0$) (SR, image) | $98.0 \pm 1.6$ | $59.3 \pm 6.8$ | $74.0 \pm 3.3$ |
| Threading ($D_1$) (SR, image) | $60.7 \pm 2.5$ | $43.3 \pm 9.3$ | - |
| Threading ($D_2$) (SR, image) | $38.0 \pm 3.3$ | $22.7 \pm 0.9$ | - |
| Square ($D_0$) (SR, low-dim) | $98.0 \pm 1.6$ | $82.0 \pm 1.6$ | $60.7 \pm 3.4$ |
| Square ($D_1$) (SR, low-dim) | $80.7 \pm 3.4$ | $70.0 \pm 1.6$ | - |
| Square ($D_2$) (SR, low-dim) | $58.7 \pm 1.9$ | $55.3 \pm 1.9$ | - |
| Threading ($D_0$) (SR, low-dim) | $97.3 \pm 0.9$ | $69.3 \pm 0.9$ | $34.7 \pm 6.6$ |
| Threading ($D_1$) (SR, low-dim) | $72.0 \pm 1.6$ | $56.7 \pm 5.0$ | - |
| Threading ($D_2$) (SR, low-dim) | $60.7 \pm 6.2$ | $46.0 \pm 7.5$ | - |

Table M.1: **Effect of Action Noise.** MimicGen adds Gaussian noise to actions when executing transformed segments during data generation. These results show that removing the noise can increase the data generation rate (as expected), but can cause agent performance to decrease significantly. They also show that just replaying the same task instances from the source human data with action noise is not sufficient (although it does improve results over just using the source human data).

The **per-subtask** hyperparameter determines how frequently source demonstration selection occurs — we next discuss strategies for actually selecting the source demonstration.

**Selection Strategy.** We now turn to how the source demonstration $j$ is selected. We found **random** selection to be a simple and effective strategy in many cases — here, we simply select the source demonstration $j$ uniformly at random from $\{1, 2...., N\}$. We used this strategy for most of our tasks. However, we found some tasks benefit from a **nearest-neighbor** selection strategy. Consider selecting a source demonstration segment for subtask $S_i(o_{S_i})$. We compare the pose $T_W^{O'_0}$ of object $o_{S_i}$ in the current scene with the initial object pose $T_W^{O_0}$ at the start of each source demonstration segment $\tau_i^j$, and sort the demonstrations (ascending) according to the pose distance (to evaluate the pose distance for each demonstration segment, we sum the $L_2$ position distance with the angle value of the delta rotation (in axis-angle form) between the two object rotations). We then select a demonstration uniformly at random from the first $nn_k$ members of the sorted list.

## M.4 Action Noise

When MimicGen executes a transformed segment during data generation, it converts the sequence of target poses into delta-pose actions $a_t$ at each timestep. We found it beneficial to apply additive noise to these actions — we apply Gaussian noise $\mathcal{N}(0, 1)$ with magnitude $\sigma$ in each dimension (excluding gripper actuation). To showcase the value of including the noise we ran an ablation experiment (presented in Table M.1) that shows how much data generation success rate and agent performance changes when the datasets are not generated with action noise during execution (compared to our default value of $\sigma = 0.05$).

As expected, the data generation success rate increases when using no noise, as noise can cause the end effector motion to deviate from the expected subtask segment that is being followed (the most significant example is an increase of 33% on Threading $D_0$). However, agent performance suffers, with performance drops as large as 30% on agents trained on low-dim observations, and up to 40% on agents trained on image observations.

Another natural question is whether the benefits of MimicGen come purely from action noise injection. To investigate this, we also ran a comparison ("replay w/ noise" in Table M.1) where we took the 10 source demos, and replayed them with the same level of action noise (0.05) used in our experiments until we collected 1000 successful demonstrations. We selected a random source

| Task | normal | no NN | no per-subtask | no NN + no per-subtask |
|---|---|---|---|---|
| Square ($D_0$) (DGR) | 73.7 | 36.7 | - | - |
| Square ($D_1$) (DGR) | 48.9 | 30.6 | - | - |
| Square ($D_2$) (DGR) | 31.8 | 22.4 | - | - |
| Nut Assembly ($D_0$) (DGR) | 50.0 | 27.1 | - | - |
| Stack ($D_0$) (DGR) | 94.3 | - | 85.1 | 71.6 |
| Stack ($D_1$) (DGR) | 90.0 | - | 76.3 | 63.3 |
| Stack Three ($D_0$) (DGR) | 71.3 | - | 37.8 | 26.7 |
| Stack Three ($D_1$) (DGR) | 68.9 | - | 36.0 | 27.5 |
| Pick Place ($D_0$) (DGR) | 32.7 | - | 30.8 | 29.7 |
| Square ($D_0$) (SR, low-dim) | $98.0 \pm 1.6$ | $94.7 \pm 2.5$ | - | - |
| Square ($D_1$) (SR, low-dim) | $80.7 \pm 3.4$ | $79.3 \pm 2.5$ | - | - |
| Square ($D_2$) (SR, low-dim) | $58.7 \pm 1.9$ | $57.3 \pm 0.9$ | - | - |
| Nut Assembly ($D_0$) (SR, low-dim) | $76.0 \pm 1.6$ | $64.7 \pm 5.7$ | - | - |
| Stack ($D_0$) (SR, low-dim) | $100.0 \pm 0.0$ | - | $99.3 \pm 0.9$ | $99.3 \pm 0.9$ |
| Stack ($D_1$) (SR, low-dim) | $100.0 \pm 0.0$ | - | $100.0 \pm 0.0$ | $99.3 \pm 0.9$ |
| Stack Three ($D_0$) (SR, low-dim) | $88.0 \pm 1.6$ | - | $84.0 \pm 1.6$ | $81.3 \pm 2.5$ |
| Stack Three ($D_1$) (SR, low-dim) | $90.7 \pm 0.9$ | - | $78.7 \pm 2.5$ | $83.3 \pm 0.9$ |
| Pick Place ($D_0$) (SR, low-dim) | $58.7 \pm 7.5$ | - | $52.0 \pm 3.3$ | $56.0 \pm 5.9$ |

Table M.2: **Effect of Removing Selection Strategy.** Some of our tasks used a nearest-neighbor selection strategy and a per-subtask selection strategy for source demonstration segments. These results show the effect of removing these selection strategies (e.g. using the default, random selection strategy). Interestingly, while the data generation rates decrease significantly, agent performance does not decrease significantly for most tasks.

demonstration at the start of each trial and reset the simulator state to its initial state before collection.

This comparison shows the value of using MimicGen to transform and interpolate source human segments to collect data on new configurations, instead of purely using replay with noise on the same configurations from the source data. Comparing the "replay w/ noise" column of Table M.1 to Fig. 4, we see that there is an appreciable increase in the success rate on $D_0$ compared to just using the 10 source demos (Square increases from 11.3 to 42.0, and Threading increases from 19.3 to 74.0), but training on the MimicGen dataset still achieves better performance on $D_0$ (Square: 90.7, Threading: 98).

## M.5 Data Generation for Mobile Manipulation Tasks

The process of transforming source segments differs slightly for mobile manipulation tasks. A source segment may or may not contain mobile base actions. If the segment does not contain mobile base actions we generate segments in the same manner as our method for manipulator-only environments. If a segment does contain mobile base actions we assume that the segment can be split into three contiguous sub-segments: (1) a sub-segment involving manipulator actions, (2) a subsequent sub-segment involving mobile base actions, and (3) a final sub-segment involving manipulator actions. We generate corresponding sub-segments for each of these phases. We generate sub-segments for (1) and (3) in the same manner as our algorithm for manipulator-only environments, and we generate sub-segment (2) by simply copying the mobile base actions from the reference sub-segment. We found this scheme to work sufficiently well for the mobile manipulation task in this work, but future work improve the generation of sub-segment (2) (the robot base movement) to account for different environment layouts in a scene, by defining and using a reference frame for each base motion segment, like the object-centric subtasks used for arm actions, and/or integrating a motion planner for the base. We highlight the limitations of our approach in Appendix C.

## M.6 MimicGen Hyperparameters

In this section, we summarize the data generation hyperparameters (defined above) used for each task. As several tasks had the same settings, we group tasks together wherever possible.

**Default.** Most of our tasks used a noise scale of $\sigma = 0.05$, interpolation steps of $n_{\text{interp}} = 5$, $n_{\text{fixed}} = 0$, and a selection strategy of **random** with **per-subtask** set to False. These tasks include

Threading, Coffee, Three Piece Assembly, Hammer Cleanup, Mug Cleanup, Kitchen, Coffee Preparation, Mobile Kitchen, Nut-and-Bolt Assembly, Gear Assembly, and Frame Assembly.

**Nearest-Neighbor and Per-Subtask.** Some of our tasks used the default values above, with the exception of using a **nearest-neighbor** selection strategy. The following tasks used **nearest-neighbor** ($nn_k = 3$) with **per-subtask** set to False: Square and Nut Assembly. Some tasks used **nearest-neighbor** ($nn_k = 3$) with **per-subtask** set to True: Stack, Stack Three, Pick Place. In general, we found **per-subtask** selection to help for pick-and-place tasks. To showcase the value of using these specific selection strategies, we ran an ablation experiment (presented in Table M.2) that shows how much data generation success rate and agent performance changes when turning these strategies off during data generation. Interestingly, while the data generation rates decrease significantly, agent performance does not decrease significantly for most tasks.

**Real.** Our real robot tasks used different settings for safety considerations, and to ensure that data could be collected in a timely manner (maintain high data generation rate). All tasks used a reduced noise scale of $\sigma = 0.02$, and higher interpolation steps of $n_{\text{interp}} = 25$, $n_{\text{fixed}} = 25$. The Stack task used a selection strategy of **nearest-neighbor** ($nn_k = 3$) with **per-subtask** set to True, and the Coffee task used a selection strategy of **random** with **per-subtask** set to False, just like their simulation counterparts.

# N  Policy Training Details

We describe details of how policies were trained via imitation learning. Several design choices are the same as the robomimic study [7].

**Observation Spaces.** As in robomimic [7], we train policies on two observation spaces — "low-dim" and "image". While both include end effector poses and gripper finger positions, "low-dim" includes ground-truth object poses, while "image" includes camera observations from a front-view camera and a wrist-view camera. All tasks use images with 84x84 resolution with the exception of the real world tasks (Stack, Coffee), which use an increased resolution of 120x160. For "image" agents, we apply pixel shift randomization [7, 36–39] and shift image pixels by up to 10% of each dimension each time observations are provided to the agent.

**Training Hyperparameters.** We use BC-RNN from robomimic [7] with the default hyperparameters reported in their study, with the exception of an increased learning rate (1e-3 instead of 1e-4) for policies trained on low-dim observations, as we found it to speed up policy convergence on large datasets.

**Policy Evaluation.** As in [7], on simulation tasks, we evaluate policies using 50 rollouts per agent checkpoint during training, and report the maximum success rate achieved by each agent across 3 seeds. On the real world tasks, due to the time-consuming nature of policy evaluation, we take the last policy checkpoint produced during training, and evaluate it over 50 episodes.

**Hardware.** Each data generation run and training run used a machine (on a compute cluster) with an NVIDIA Volta V100 GPU, 8 CPUs, 32GB of memory, and 128GB of disk space. In certain cases, we batched multiple data generation runs and training runs on the same machine (usually 2 to 4 runs). Real robot experiments were carried out on a machine with an NVIDIA GeForce RTX 3090 GPU, 36 CPUs, 32GB of memory, and 1 TB of storage.

# O  Data Generation Success Rates

In this section, we present data generation success rates for each of our generated datasets. Comparing the results in Table O.1 with our core image-based agent results (Fig. 4) and low-dim agent results (Table P.1), we see that in many cases the agent performance is much higher than the data generation success rate. An extreme example is the Gear Assembly task which has data generation rates of $46.9\%$ ($D_0$), $8.2\%$ ($D_1$), and $7.1\%$ ($D_2$) but policy success rates of $92.7\%$ ($D_0$), $76.0\%$ ($D_1$), and $64.0\%$ ($D_2$). We also saw much higher agent performance than the data generation rate in our robot transfer experiment (see Appendix E).

| Task | $D_0$ | $D_1$ | $D_2$ |
|---|---|---|---|
| Stack | 94.3 | 90.0 | - |
| Stack Three | 71.3 | 68.9 | - |
| Square | 73.7 | 48.9 | 31.8 |
| Threading | 51.0 | 39.2 | 21.6 |
| Coffee | 78.2 | 63.5 | 27.7 |
| Three Pc. Assembly | 35.6 | 35.5 | 31.3 |
| Hammer Cleanup | 47.6 | 20.4 | - |
| Mug Cleanup | 29.5 | 17.0 | - |
| Kitchen | 100.0 | 42.7 | - |
| Nut Assembly | 50.0 | - | - |
| Pick Place | 32.7 | - | - |
| Coffee Preparation | 53.2 | 36.1 | - |
| Mobile Kitchen | 20.7 | - | - |
| Nut-and-Bolt Assembly | 66.0 | 59.4 | 47.6 |
| Gear Assembly | 46.9 | 8.2 | 7.1 |
| Frame Assembly | 45.3 | 32.7 | 28.9 |

Table O.1: **Data Generation Rates.** For each task that we generated data for, we report the data generation rate (DGR) — which is the success rate of the data generation process (recall that not all data generation attempts are successful, and MimicGen only keeps the attempts that result in task success). Comparing with Table P.1 and Fig. 4, we can see that several tasks have significantly higher policy learning performance than data generation rates.

# P   Low-Dim Policy Training Results

1345  In the main text we focused on *image* observation spaces. In this section we present full results
1346  for agents trained on *low-dim* observation spaces and show that these agents are equally perfor-
1347  mant. Results on our main generated datasets are shown in Table P.1 (and can be compared to the
1348  image-based agent results in Fig. 4), and the source dataset size comparison and policy training data
1349  comparisons are shown in Fig. P.1 (and can be compared to Fig. 4).

| Task | Source | $D_0$ | $D_1$ | $D_2$ |
|---|---|---|---|---|
| Stack | $38.7 \pm 4.1$ | $100.0 \pm 0.0$ | $100.0 \pm 0.0$ | - |
| Stack Three | $2.7 \pm 0.9$ | $88.0 \pm 1.6$ | $90.7 \pm 0.9$ | - |
| Square | $18.7 \pm 0.9$ | $98.0 \pm 1.6$ | $80.7 \pm 3.4$ | $58.7 \pm 1.9$ |
| Threading | $9.3 \pm 2.5$ | $97.3 \pm 0.9$ | $72.0 \pm 1.6$ | $60.7 \pm 6.2$ |
| Coffee | $42.7 \pm 4.1$ | $100.0 \pm 0.0$ | $93.3 \pm 2.5$ | $76.7 \pm 0.9$ |
| Three Pc. Assembly | $2.7 \pm 0.9$ | $74.7 \pm 3.8$ | $61.3 \pm 1.9$ | $38.7 \pm 4.1$ |
| Hammer Cleanup | $64.7 \pm 4.1$ | $100.0 \pm 0.0$ | $74.0 \pm 1.6$ | - |
| Mug Cleanup | $8.0 \pm 1.6$ | $82.0 \pm 2.8$ | $54.7 \pm 5.0$ | - |
| Kitchen | $43.3 \pm 3.4$ | $100.0 \pm 0.0$ | $78.0 \pm 2.8$ | - |
| Nut Assembly | $0.0 \pm 0.0$ | $76.0 \pm 1.6$ | - | - |
| Pick Place | $0.0 \pm 0.0$ | $58.7 \pm 7.5$ | - | - |
| Coffee Preparation | $2.0 \pm 0.0$ | $76.0 \pm 5.7$ | $59.3 \pm 3.4$ | - |
| Mobile Kitchen | $6.7 \pm 3.8$ | $76.7 \pm 10.5$ | - | - |
| Nut-and-Bolt Assembly | $2.0 \pm 0.0$ | $98.0 \pm 1.6$ | $96.0 \pm 1.6$ | $81.3 \pm 3.8$ |
| Gear Assembly | $12.0 \pm 1.6$ | $92.7 \pm 1.9$ | $76.0 \pm 4.9$ | $64.0 \pm 3.3$ |
| Frame Assembly | $9.3 \pm 3.4$ | $87.3 \pm 2.5$ | $70.7 \pm 1.9$ | $58.0 \pm 5.7$ |

Table P.1: **Low-Dim Agent Performance on Source and Generated Datasets.** For each task, we present the
success rates (3 seeds) of low-dim agents trained with BC on the 10 source demos and on each MimicGen
dataset (1000 demos for each reset distribution). There is a large improvement across all tasks on the default
distribution ($D_0$) and agents are performant on the broader distributions ($D_1$, $D_2$).

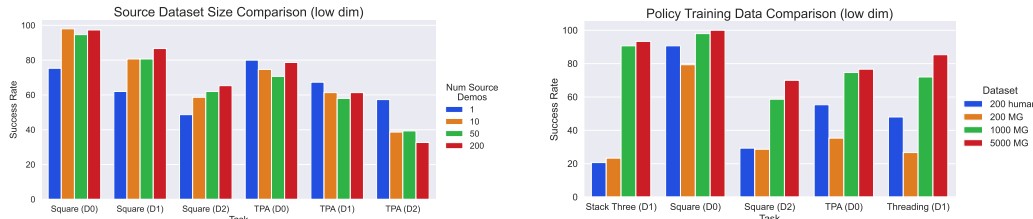

Figure P.1: (left) **MimicGen with more source human demonstrations.** We found that using larger source
datasets to generate MimicGen data did not result in significant low-dim agent improvement. (right) **Policy
Training Dataset Comparison.** We compare agents trained on 200 MimicGen demos to 200 human demos —
remarkably, the performance is similar, despite MimicGen only using 10 source human demos. MimicGen can
also produce improved low-dim agents by generating datasets — we show a comparison between 200, 1000,
and 5000 above. However, there can be diminishing returns.

## Q   Bias and Artifacts in Generated Data

In this section, we discuss some undesirable properties of the generated data.

**Are datasets generated by MimicGen biased towards certain scene configurations?** This is a natural question to ask, since MimicGen keeps trying to re-use the same small set of human demonstrations on new scenes and only retains the successful traces. Indeed, there might be a limited set of scene configurations where data generation works successfully, and some scene configurations that are never included in the generated data. We conduct an initial investigation into whether such bias exists by analyzing the set of initial states in a subset of our generated datasets. Specifically, we take inspiration from [78], and discretize the set of possible object placements for each object in each task into bins. Then, we simply maintain bin counts by taking the initial object placements for each episode in a generated dataset, computing the bin it belongs to, and updating the bin count. Finally, we estimate the *support coverage* of the reset distribution by counting the number of non-zero bins and dividing by the total number of bins.

As a concrete example, consider the Threading $D_1$ variant, where the needle and tripod are both sampled from a region with bounds in $x$, $y$ and $\theta$, where $\theta$ is a top-down rotation angle (see Fig. 5). If each dimension is discretized into $n$ independent bins, there are a total of $n^6$ bins (all combinations of the dimensions). Due to this exponential scaling, we use a small number of bins ($n = 3$). Note that when conducting this analysis, we had to be careful to ensure that the overall bin count was not too small or too large. If it was too small, each bin would correspond to a large section of the object configuration space, and the results would not be meaningful. Similarly, if it was too large, there is no way for 1000 generated demonstrations to cover a meaningful portion of the support (since there can only be 1000 bins covered at best).

We now present our results. For several environments, we found there to be a good amount of support coverage — for example, Coffee $D_1$ (98.8%), Coffee $D_2$ (89.3%), and Square $D_1$ (92.6%). However, we also found datasets that likely have significant amounts of bias — for example, Square $D_2$ (66.4%), Threading $D_1$ (71%), Threading $D_2$ (61.2%), Three Piece Assembly $D_0$ (67.9%), Three Piece Assembly $D_1$ (43.5%), and Mug Cleanup $D_1$ (64%). This analysis is certainly imperfect, as some datasets could still be biased towards containing certain object configurations than others (e.g. having non-uniform bin counts across the support), and there could also be different kinds of bias (such as repetitive motions). However, this analysis does confirm that there is certainly bias in some of the generated datasets. A deeper investigation into the properties of the generated data is left for future work.

**Are there artifacts and other undesirable behavior characteristics in MimicGen datasets?** Artifacts and other undesirable behavior characteristics are likely, for two reasons. One reason is that MimicGen bridges transformed segments from the source dataset with interpolation segments. These interpolation segments could result in long paths and unnatural motions that are difficult to imitation. In fact, we found some evidence of this fact (see Appendix G). Another reason is that MimicGen only checks for a successful task completion when deciding whether to accept a generated trajectory. This means that there might be undesirable behaviors such as collisions between the robot and certain parts of the world (including objects that are not task-relevant). As we move towards deploying robots trained through imitation learning, data curation efforts are of the utmost importance — this is left for future work.

 # R    Using More Varied Source Demonstrations

| Task | Source | $D_0$ | $D_1$ | $D_2$ |
|---|---|---|---|---|
| Square (src $D_0$) (DGR) | - | 73.7 | 48.9 | 31.8 |
| Square (src $D2$) (DGR) | - | 54.4 | 51.7 | 52.3 |
| Three Piece Assembly (src $D_0$) (DGR) | - | 35.6 | 35.5 | 31.3 |
| Three Piece Assembly (src $D_2$) (DGR) | - | 26.9 | 29.1 | 23.9 |
| Square (src $D_0$) (SR, low-dim) | $18.7 \pm 0.9$ | $98.0 \pm 1.6$ | $80.7 \pm 3.4$ | $58.7 \pm 1.9$ |
| Square (src $D_2$) (SR, low-dim) | $2.0 \pm 0.0$ | $98.0 \pm 1.6$ | $84.7 \pm 1.9$ | $60.7 \pm 2.5$ |
| Three Piece Assembly (src $D_0$) (SR, low-dim) | $2.7 \pm 0.9$ | $74.7 \pm 3.8$ | $61.3 \pm 1.9$ | $38.7 \pm 4.1$ |
| Three Piece Assembly (src $D_2$) (SR, low-dim) | $0.0 \pm 0.0$ | $62.0 \pm 4.9$ | $57.3 \pm 4.1$ | $32.0 \pm 2.8$ |

Table R.1: **Using More Varied Source Demonstrations.** We present a comparison of data generation success rates and policy success rates (3 seeds) across two choices of source datasets — the 10 source human demonstrations collected on $D_0$ (default used in main experiments) and 10 source human demonstrations collected on the significantly more diverse $D_2$ reset distribution. Interestingly, while the data generation success rates differ, the policy success rates are comparable, suggesting that downstream agent performance can be invariant to how much the task initializations of the source demonstrations vary.

Most of our experiments used 10 source human demonstrations collected on a narrow reset distribution ($D_0$) and generated demonstrations with MimicGen across significantly more varied reset distributions ($D_0$, $D_1$, $D_2$). In this section, we investigate whether having source demonstrations collected on a more varied set of task initializations is helpful. We do this by collecting 10 source human demonstrations on $D_2$ and using it to generate data for all reset distributions ($D_0$, $D_1$, $D_2$). The results are presented in Table R.1. Interestingly, while the data generation success rates differ, the policy success rates are comparable, suggesting that downstream agent performance can be invariant to how much the task initializations of the source demonstrations vary.

# S  Data Generation with Multiple Seeds

MimicGen's data generation process has several sources of randomness, including the initial state of objects for each data generation attempt (which is sampled from the reset distribution $D$), selecting the source dataset segment that will be transformed (Appendix M.3), and the noise added to actions during execution (Appendix M.4). In all of our experiments, we only used a single seed to generate datasets (our policy learning results are reported across 3 seeds though). In this section, we justify this decision, by showing that there is very little variance in empirical results across different data generation seeds.

We generated 3 datasets (3 different seeds) for Stack Three ($D_0$, $D_1$) and Square ($D_0$, $D_1$, $D_2$), and train low-dim policies (3 seeds per generated results, so 9 seeds in total per task variant) and summarize the results in Table S.1. The data generation success rates have very tight variance (less than 1%) and do not deviate from our reported data generation rates (Appendix O) by more than 0.6%. Furthermore, the mean policy success rates are extremely close to our reported results for low-dim agents in Table P.1 (less than 2% deviation).

| Task | $D_0$ | $D_1$ | $D_2$ |
|---|---|---|---|
| Stack Three (DGR) | $71.7 \pm 0.3$ | $69.3 \pm 0.4$ | - |
| Square (DGR) | $74.4 \pm 0.5$ | $48.5 \pm 0.7$ | $32.0 \pm 0.9$ |
| Stack Three (SR) | $89.6 \pm 2.1$ | $92.4 \pm 1.6$ | - |
| Square (SR) | $96.7 \pm 2.1$ | $81.6 \pm 4.5$ | $58.0 \pm 3.5$ |

Table S.1: **Data Generation with Multiple Seeds.** We present data generation rates (DGR) and success rates (SR) across 3 seeds of data generation, and 3 low-dim policy training seeds per dataset (9 seeds) total. The results are very close to our reported results (less than 0.6% deviation in DGR, less than 2% deviation in SR) despite our results only generating datasets with one seed.

# T    Tolerance to Pose Estimation Error

In the main text, we demonstrated that MimicGen is fully functional in real-world settings and can operate with minimal assumptions (e.g. no special tags or pose trackers) by using pose estimation methods (see Appendix G for details). Consequently, the data generation process has some tolerance to pose error and can operate without having access to perfect pose estimates. In this section, we further investigate this tolerance in simulation by adding 2 levels of uniform noise to object poses - L1 is 5 mm position and 5 deg rotation noise and L2 is 10 mm position and 10 deg rotation noise [107]. As shown in Table T.1, the data generation rate decreases (e.g. Square D0 decreases from 73.7% to 60.9% for L1 and 30.5% for L2 and Square D2 decreases from 31.8% to 25.1% for L1 and 14.5% for L2), but visuomotor policy learning results are relatively robust (Square D0 decreases from 90.7% to 89.3% for L1 and 84.7% for L2, and Square D2 decreases from 49.3% to 47.3% for L1 and 39.3% for L2).

| Task | None | Level 1 (5 mm / 5 deg) | Level 2 (10 mm / 10 deg) |
|---|---|---|---|
| Stack Three ($D_1$) (DGR) | 68.9 | 62.3 | 38.7 |
| Stack Three ($D_1$) (SR) | $86.7 \pm 3.4$ | $84.0 \pm 2.8$ | $80.7 \pm 3.4$ |
| Square ($D_0$) (DGR) | 73.7 | 60.9 | 30.5 |
| Square ($D_1$) (DGR) | 48.9 | 40.2 | 20.2 |
| Square ($D_2$) (DGR) | 31.8 | 25.1 | 14.5 |
| Square ($D_0$) (SR) | $90.7 \pm 1.9$ | $89.3 \pm 2.5$ | $84.7 \pm 2.5$ |
| Square ($D_1$) (SR) | $73.3 \pm 3.4$ | $64.0 \pm 1.6$ | $62.0 \pm 1.6$ |
| Square ($D_2$) (SR) | $49.3 \pm 2.5$ | $47.3 \pm 6.8$ | $39.3 \pm 4.7$ |
| Coffee ($D_0$) (DGR) | 78.2 | 28.9 | 5.6 |
| Coffee ($D_1$) (DGR) | 63.5 | 22.6 | 4.3 |
| Coffee ($D_0$) (SR) | $100.0 \pm 0.0$ | $95.3 \pm 2.5$ | $79.3 \pm 0.9$ |
| Coffee ($D_1$) (SR) | $90.7 \pm 2.5$ | $83.3 \pm 2.5$ | $77.3 \pm 4.1$ |
| Threading ($D_0$) (DGR) | 51.0 | 17.6 | 5.2 |
| Threading ($D_0$) (SR) | $98.0 \pm 1.6$ | $94.7 \pm 0.9$ | $86.7 \pm 1.9$ |

Table T.1: **Tolerance to Noisy Pose Estimates.** We investigate how the data generation success rates (DGR) and visuomotor policy success rates (SR) change when adding uniform pose noise to the object poses in the source demonstrations and the new scene during data generation. Although the data generation rates decrease, policy success rates are robust. This shows that MimicGen can be tolerant to noisy object pose estimation, and is suitable for real-world data collection.

