# OpenReview forum: "MimicGen: A Data Generation System for Scalable Robot Learning using Human Demonstrations"
_robot-learning.org/CoRL/2023/Workshop/TGR — CoRL 2023 Workshop TGR Oral_

### Official Review · Reviewer_bdtG · 2023-10-19

**Rating:** 8
**Confidence:** 5

**Review:**

The idea for scaling up demonstrations based on costly and limited human-provided ones is sound and the dataset is valuable. The topic also closely aligns with the workshop direction.

---

### Official Review · Reviewer_4YtR · 2023-10-19

**Rating:** 7
**Confidence:** 3

**Review:**

This work proposes MimicGen, a system for automatically synthesizing large-scale diverse dataset by adapting a small number of human demonstrations to new contexts. It can generate over 50k demonstration across 18 tasks with variants on scene configurations, object instances, and robot arms from only ~200 human demonstration. They showcase much superior agent performance using the generated data from MimicGen as opposed to learning from the smaller-sized human demonstration only. Overall, this work presents a simple yet effective method to augment human demonstration toward scalable imitation learning for robot agents and is highly relevant with the workshop topic.

---

### Decision · Program_Chairs · 2023-10-20

**Decision:**

Accept (Oral)

**Comment:**

Great paper and closely aligned topic!